# Self-recycling and partially conservative replication of mycobacterial methylmannose polysaccharides

Ana Maranha [1,2,3,8], Mafalda Costa[1,8], Jorge Ripoll-Rozada [4,5,7,8], José A. Manso [4,5], Vanessa Miranda[6], Vera M. Mendes[1,2], Bruno Manadas [1,2], Sandra Macedo-Ribeiro [4,5], M. Rita Ventura [6], Pedro José Barbosa Pereira [4,5 ✉] & Nuno Empadinhas [1,2,3 ✉]

The steep increase in nontuberculous mycobacteria (NTM) infections makes understanding their unique physiology an urgent health priority. NTM synthesize two polysaccharides proposed to modulate fatty acid metabolism: the ubiquitous 6-O-methylglucose lipopolysaccharide, and the 3-O-methylmannose polysaccharide (MMP) so far detected in rapidly growing mycobacteria. The recent identification of a unique MMP methyltransferase implicated the adjacent genes in MMP biosynthesis. We report a wide distribution of this gene cluster in NTM, including slowly growing mycobacteria such as *Mycobacterium avium*, which we reveal to produce MMP. Using a combination of MMP purification and chemoenzymatic syntheses of intermediates, we identified the biosynthetic mechanism of MMP, relying on two enzymes that we characterized biochemically and structurally: a previously undescribed α–endomannosidase that hydrolyses MMP into defined-sized mannoligosaccharides that prime the elongation of new daughter MMP chains by a rare α-(1→4)-mannosyltransferase. Therefore, MMP biogenesis occurs through a partially conservative replication mechanism, whose disruption affected mycobacterial growth rate at low temperature.

[1] CNC - Center for Neuroscience and Cell Biology, University of Coimbra, 3004-504 Coimbra, Portugal. [2] CIBB - Center for Innovative Biomedicine and Biotechnology, University of Coimbra, Coimbra, Portugal. [3] IIIUC - Institute of Interdisciplinary Research, University of Coimbra, 3030-789 Coimbra, Portugal. [4] IBMC - Instituto de Biologia Molecular e Celular, Universidade do Porto, 4200-135 Porto, Portugal. [5] Instituto de Investigação e Inovação em Saúde, Universidade do Porto, 4200-135 Porto, Portugal. [6] Bioorganic Chemistry Group, Instituto de Tecnologia Química Biológica António Xavier, Universidade Nova de Lisboa (ITQB NOVA), Av. da República, 2780-157 Oeiras, Portugal. [7] Present address: Instituto de Biomedicina y Biotecnología de Cantabria (IBBTEC), Consejo Superior de Investigaciones Científicas (CSIC)-Universidad de Cantabria, Santander, Spain. [8] These authors contributed equally: Ana Maranha, Mafalda Costa, Jorge Ripoll-Rozada. ✉email: ppereira@ibmc.up.pt; numenius@cnc.uc.pt

Nontuberculous mycobacteria (NTM) thrive in soils and natural waters and have often been isolated from household water distribution systems[1]. Some species are opportunistic pathogens causing serious infections in the lungs, skin, bones, joints or in the central nervous system[2]. Chronic exposure is an obvious risk factor, but immune fragilities, pre-existing lung disease or structural abnormalities are also known to increase susceptibility to NTM infection[3]. The clinical presentation of NTM lung infection can be similar to that of tuberculosis, and in cases of extrapulmonary infection, the disease assumes a range of nonspecific manifestations[4,5]. The inexistence of rapidly acting antibiotics to fight NTM infections results in lengthy and poorly tolerated drug regimens often accompanied by severe side-effects, making the infection prone to relapse and drug resistance to accumulate[6]. The need for better therapeutic schemes directed at each NTM species, and according to their drug susceptibility patterns, represent enormous challenges which, adding to their extraordinary environmental resilience, further curbs the control of NTM disease worldwide[5]. The increasing incidence of NTM lung disease calls for further efforts in identifying pathways that can afford new drug targets and more effective therapies.

The mycobacterial cell envelope, a coat with intricate layers of sugars, peptides and lipids, makes these organisms extremely resilient while also contributing to their interactions with the invaded host and pathogenicity[7]. Mycobacteria synthesize intracellular polymethylated polysaccharides (PMPS) of 6-O-methyl-glucose (MGLP) and of 3-O-methylmannose (MMP), both proposed to sequester and protect newly formed fatty-acyl CoA chains in the cytoplasm, thus modulating fatty acid (FA) metabolism and indirectly the lipidic cell envelope[8,9]. While MGLP has been isolated from all mycobacteria examined thus far[8,9] and implicated in adaptation to heat stress[10], MMP has only been extracted from some rapidly growing mycobacteria (RGM) where its importance under environmental stress or during infection is unknown[11,12].

MMP is composed of a linear chain of 11 to 14 units of α-(1→4)-linked 3-O-methyl-D-mannose (3-O-MeMan), blocked in the reducing end by a methyl aglycon, and with an unmethylated mannose at the nonreducing end[8]. The α-(1→4) linkage between mannoses found in MMP appears to be extremely rare in nature[13], which renders the key α-(1→4)-mannosyltransferase a functionally and structurally unique glycosyltransferase. Indeed, mannans from plants show a clear prevalence of β-(1→4) linkages, whereas those from fungi and bacteria essentially contain α-(1→2), α-(1→3), and α-(1→6) linkages and branching[14–16]. Also, in the mycobacterial mannose polymers phosphatidyl-myo-inositol mannosides (PIM), lipomannan (LM) and lipoarabinomannan (LAM), mannose units are essentially connected through α-(1→6) and α-(1→2) glycosidic bonds[7]. Sugar methylation occurs infrequently in fungi and bacteria but is a very common modification in mycobacteria[17]. Such methylated sugars appear at terminal positions, this being especially true for 3-O-MeMan that is usually found as a monomeric constituent of a glycan or at the non-reducing end where it likely acts as a stop signal of chain elongation. However, the consecutive arrangement of methylated sugars found in MMP is uncommon[17]. An exception is a lipopolysaccharide from the soil bacterium *Oligotropha carboxidovorans*, in which the O-specific polysaccharide is a homopolymer of 35-40 units of α-(1→2) linked 3-O-MeMan[16].

In addition to the mannosyltransferases, other enzymes crucial to balance the cell mannosylome are the mannoside-degrading enzymes, all grouped in the CAZY database (http://www.cazy.org)[18]. In contrast with the diversity of known mannan-degrading enzymes (β-mannanases and β-manosidases), only a few fungal and bacterial α-mannosidases with activity toward polysaccharides have been characterized[15,19,20].

MMP mannosyltransferase and methyltransferase activities were detected in *M. smegmatis* extracts[21,22]. The mannosyltransferase was found to add one mannose to short (at least 4-mer) methylmannosides[21], whereas the methyltransferase could methylate position 3 of the terminal residue of a α-(1→4) methylmannose chain[22]. Because the mannosyltransferase products served as substrates for the methyltransferase and vice versa, it was proposed that MMP elongation depended on alternate mannosylation and methylation events[22]. A subsequent model for MMP polymerization proposed that mannosylation would be independent of 3-O-methylation, and that mannose units could be added to oligomannosides regardless of methylation[23].

Recently, a 4-gene cluster encoding a 1-O-methyltransferase (MeT1), an unknown protein (OrfA), a putative 3-O-methyltransferase (pMeT3) and a putative α-(1→4)-mannosyltransferase (ManT), was identified in NTM genomes and proposed to coordinate MMP biosynthesis[24]. MeT1 was a unique methyltransferase specifically blocking the 1-OH position of a dimethylmannobiose, a potential early precursor of MMP[24]. We here reveal that ManT is a rare MMP α-(1→4)-mannosyltransferase and OrfA is a unique MMP hydrolase (MmpH). ManT transfers mannose from GDP-mannose to tri- and tetramannosides (independently of methylation state) being, to our knowledge, the first enzyme to catalyze the elongation of a polymer with α-(1→4)-linked mannoses. MmpH is a highly specific hydrolase that catalyzes the internal cleavage of MMP. The MmpH products were substrates for elongation of daughter MMP chains by ManT, and for 1-O-methylation by MeT1. We here describe a novel MMP recycling and replication mechanism in which oligomannosides originating from the hydrolysis of pre-existing MMP by MmpH are the precursors of new MMP daughter chains. This study introduces a comprehensive perspective on the biogenesis of mycobacterial MMP, exempts it from a suspected role in their growth rate, while providing clues linking MMP to cold adaptation, a predominant stress condition to which many of these bacteria are naturally exposed.

## Results

**Extensive and scattered distribution of MMP genes across mycobacteria.** The 4-gene cluster proposed to be responsible for MMP biosynthesis encodes a 1-O-methyltransferase (MeT1)[24], a putative 3-O-methyltransferase (pMeT3) and two proteins whose functions were identified in this work as a α-(1→4)-mannosyltransferase (ManT) and as a MMP endomannosidase (or MMP hydrolase, MmpH; Fig. 1A). BLAST analyses against available mycobacterial genomes confirmed that the MMP gene cluster is not restricted to rapidly growing mycobacteria (RGM) (Fig. 1B and Fig. S1A), is rather widely distributed with a very conserved arrangement (Fig. S1B) among mycobacterial genomes but absent from those of some pathogens, namely of the *M. tuberculosis* and *M. terrae* complexes and also from those of *M. leprae*, *M. koreensis*, *M. haemophilum* and *M. kansasii*. The MMP cluster was also not detected in the genomes of some RGM namely of the *M. abscessus* complex, of *M. fallax*, *M. brumae*, *M. llatzarense*, *M. augbanense* and *M. phocaicum* (Fig. 1B). The MMP gene cluster thus has a much wider distribution among mycobacterial genomes than initially suspected based on the previously prevailing paradigm that considered MMP to be restricted to RGM (Fig. 1B).

**MMP is produced by RGM and SGM.** In addition to genomic surveys, we deemed essential to examine the presence of MMP in RGM and SGM. We extracted MMP (and MGLP) from *M. smegmatis* (Fig. 1C) and from *M. hassiacum* but could not

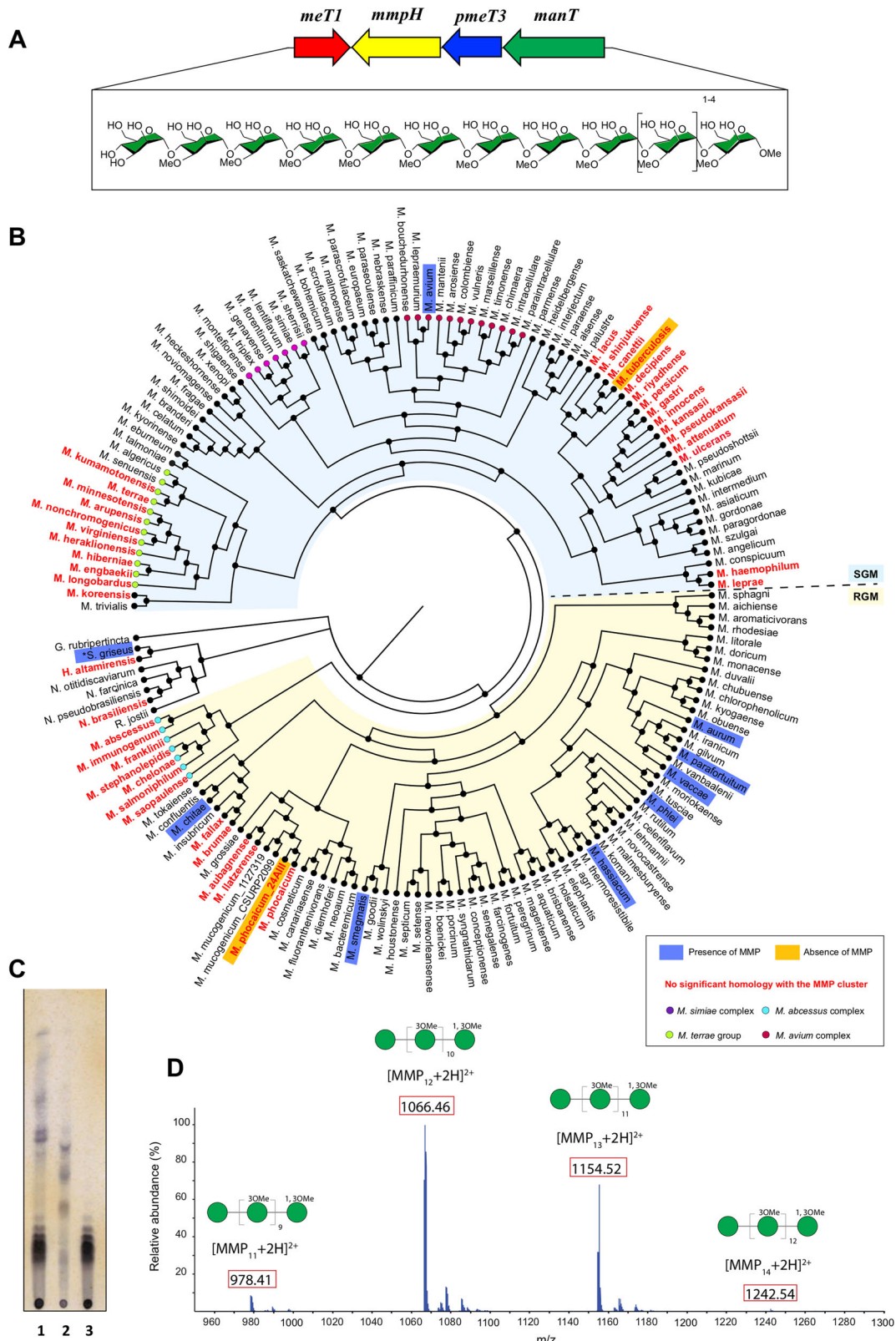

detect the polymer in an *M. phocaicum* isolate from our collection (fig. S2A). We also found that *M. avium*, an SGM containing the gene cluster, also produced MMP (Fig. 1B and Fig. S2A). Our purification strategy yielded about 3.5 mg pure MMP per liter of *M. smegmatis* culture (0.6 mg MMP/g cells). MMP variants with 11 to 14 mannose units and differential methylation, corresponding to different MMP sizes, could be assigned by mass spectrometry (MS) (Fig. 1D). The calculated MMP masses were in line with the MMP extracted from *M. smegmatis*[8,25] in which heterogeneous chains of methylated mannoses differing by one unit were isolated. The isolated MMP was composed of chains ranging in size from 11- to 14-mer, corresponding to nine to twelve 3-*O*-methylmannose units, plus a dimethylated mannose blocking the reducing end and an unmethylated mannose at the

**Fig. 1 Analysis of the presence of MMP in *Mycobacterium* and related genera. A** Organization of the cluster of MMP biosynthetic genes (arrows) in mycobacterial genomes. Red, *meT1*, 1-O-methyltransferase gene; yellow, *mmpH*, MMP hydrolase gene; blue, *pmeT3*, putative 3-O-methyltransferase gene; green, *manT*, mannosyltransferase gene. The chemical structure of MMP_11-14 and the corresponding glycan representation are given below the gene cluster scheme (boxed), according to the guidelines of Symbol Nomenclature for Glycans[80]. **B** Phylogenetic cladogram indicating the presence of the MMP gene cluster in bacterial genomes. The maximum likelihood phylogenetic tree was built based on the phylogenetic relationship between the complete genomes of 162 mycobacterial species and related actinobacteria (Table S1) as inferred using the bcgTree pipeline[44]. Previous experimental evidence of the presence of MMP was considered[10,21,81]. If not indicated in the cladogram, the strain listed in Table S1 was considered representative of the species. *Streptomyces griseus* produces an acetylated form of MMP (AMMP). **C** TLC analysis of MMP purified from *M. smegmatis* (see Methods). Lane 1, MMP-containing fractions after reversed phase chromatography; lane 2, impurities separated from the MMP sample by gel filtration chromatography; lane 3, purified MMP mixture after gel filtration chromatography. **D** ESI-TOF spectrum of the purified MMP mixture acquired in positive ion mode. A filled green circle represents one mannose residue. $[M + 2H]^{2+}$ ions are boxed red.

opposing end (Fig. S2B–D). MMP from *M. avium* and *M. hassiacum* were structurally similar (Fig. S2E–I). The nature of the glycosidic linkage and the positions of the methyl groups could not be identified from the limited structural information obtained from MS analysis and were tentatively assigned based on previously published structures[8,23,25].

**MmpH is an α-(1→4)-endomannosidase specific for the hydrolysis of MMP.** Upon confirmation of its specific glycosyl hydrolase activity toward MMP, described below, the protein of unknown function detected in the MMP gene cluster[24] (previously OrfA) was renamed MmpH (MMP hydrolase) (Fig. 2A). MmpH had low sequence identity (<30%) with α-mannosidases, β-mannanases and with other glycosyl hydrolases (Fig. S3 and Table S5). Recombinant MmpH was able to hydrolyze purified MMP into distinct lower order oligomannosides (Fig. 2A–C) but did not cleave acylated or deacylated forms of MGLP, β-mannans, synthetic 4α-oligomannosides, or its own reaction products (Fig. S4A). Purified MmpH products were identified by MS as four distinct oligomannosides (**a** to **d**) differing in the number of mannose units (4 to 8) and methylation pattern (free or methylated C1-OH) (Fig. 2A, C, D and Fig. S4B, C). The masses of products **a** and **b** were compatible with those of linear chains of four or five mannoses, respectively, all methylated at C3 and at the terminal C1 position (Fig. 2D). Likewise, the masses of products **c** and **d** were compatible with chains of seven or eight mannoses, respectively, with all but one unit methylated at position 3 (Fig. 2D). Because MS/MS fragmentation was uninformative to assign the methylation positions, those were inferred from the positions proposed previously[8]. The masses and deduced structures of the four MmpH oligomannoside products are consistent with the cleavage occurring internally in the MMP backbone, with products **a** and **b** retaining the MMP dimethylated end and products **c** and **d** matching the segment with the unmethylated end (Fig. 2A, D). These observations support the classification of MmpH as an α-(1→4)-endomannosidase. The inability to purify a homogenously sized MMP hampered distinguishing if the MmpH cleavage products result from the preferential processing of 12-mer MMP (MMP_12) with random cleavage at two distinct sites, or from the cleavage of substrates with varying lengths (MMP_11-14). In reactions ranging from 5 min to 2 h, the first product to be formed was likely **b** (pentamannoside), with an apparent formation rate more pronounced than that of product **a** (tetramannoside) (Fig. 2B), suggesting that production of **b** is favored over that of **a** in vitro and possibly in vivo. The migration patterns of **c** and **d** were indiscernible from that of MMP (Fig. 2C), impairing a similar analysis for these compounds. However, octamannoside **d** was consistently purified at higher levels than heptamannoside **c**, suggesting preferential production of **b** and **d** by MmpH.

MmpH displayed activity toward the MMPs from *M. hassiacum* and *M. avium*, underscoring its general ability to hydrolyze mycobacterial MMP (Fig. S4A).

**Biochemical and kinetic properties of MmpH.** The divalent cation-independent activity of recombinant *M. hassiacum* MmpH was detectable between 25 and 65 °C, peaking at 45 °C (Fig. 2E), with optimum pH at 6.5 (Fig. 2F). Unsurprisingly, given the thermophilic nature of its native host, MmpH was more efficient at 45 °C than at 37 °C, with comparable apparent $K_m$ and higher calculated $V_{max}$ (Fig. 2G, H and Table 1).

**The products of MmpH activity are substrates for MeT1.** MeT1 was previously shown to specifically methylate the 1-OH position of a synthetic 3-O-methylated dimannoside (sMetMan_2), a proposed initial precursor of MMP[24]. MeT1 was also able to methylate the natural MmpH products, heptamannoside **c** and octamannoside **d**, but not tetramannoside **a** and pentamannoside **b**, both naturally blocked with a methyl group at C1 (Fig. S5). The activity of MeT1 toward the synthetic substrate sMetMan_2 and the two natural substrates **c** and **d** was comparable, although with an apparent slight preference for the octamannoside **d** (Fig. 2I).

**Overall structure of MmpH.** The three-dimensional structure of MmpH was determined at 1.35 Å resolution from monoclinic (space group P2₁) crystals. Although there are two MmpH molecules in the asymmetric unit of the crystals, the enzyme behaves as a monomer in solution, as suggested both by size exclusion chromatography and DLS analysis (Fig. S6), and underscored by assessment of the protein contacts in the crystal using PISA[26]. The structures of the two MmpH protomers (termed molecule A and B) are virtually identical, with a r.m.s.d. of 0.10 Å for all atoms (0.09 Å for main-chain atoms), and therefore only molecule A will be described below. MmpH is globular and mostly α-helical, with a diameter of approximately 45 Å. Thirteen α-helices (αA-αM) are arranged into two concentric layers to form an (α/α)₆-barrel. In addition, 10 short β-strands (β1-β10) form three antiparallel β-sheets - β1-β2 in β-hairpin arrangement, β3-β5 with β4 and β5 forming a β-hairpin, and β6-β10 - decorating the (α/α)₆-barrel (Fig. 3A, B). At the center of the (α/α)₆-barrel there is a deep cavity, opening to the exterior at one of the ends of the barrel. This notorious cleft is highly conserved among MmpH homologues from several genera of Actinobacteria, including *Mycobacterium*, *Corynebacterium*, *Rhodococcus*, *Streptomyces*, and *Nocardia* (Fig. 3C and Fig. S7). Importantly, this cavity is lined by the most conserved surface residues of MmpH and has an acidic nature (Fig. 3D), making it suitable to accommodate MMPs.

Despite extensive efforts, we were unable to obtain a structure of the complex between MmpH and either purified MMP or synthetic mannosides, a task further complicated by the non-homogeneity and scarcity of the former and the commercial unavailability of the later ligands. In order to gain insights into the molecular determinants of MmpH activity, we performed a structural homology search using the Dali protein structure comparison server[27], which unsurprisingly identified several (α/α)₆-barrel glucoside hydrolases or glucosyl

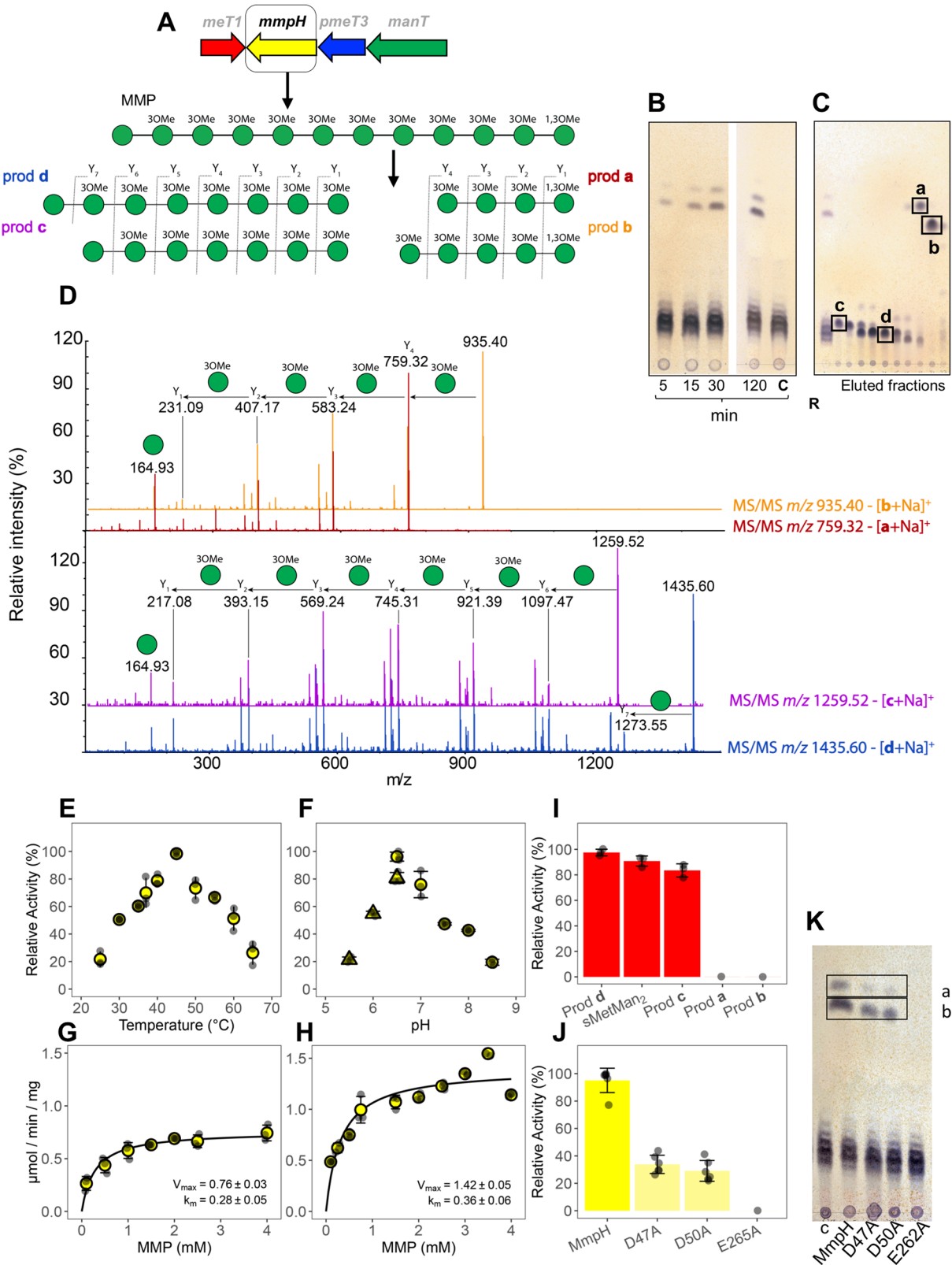

transferases. The top-scoring hits were the glucodextranase G1d from *Arthrobacter globiformis* (PDB entry 1ULV[28], Z-score = 29.9, r.m.s.d. = 3.3 Å, 13% amino acid sequence identity, glycoside hydrolase family 15 (GH15)) and two exo α-(1→6)-mannosidases belonging to family GH125: SpGH125 from *Streptococcus pneumoniae* (PDB entry 3QRY[29], Z-score = 25.2, r.m.s.d. = 3.4 Å, 8% amino acid sequence identity) and CpGH125 from *Clostridium*

*perfringens* (PDB entry 3QT9[29], Z-score = 24.9, r.m.s.d. = 3.6 Å, 10% amino acid sequence identity). Family GH125 is a relatively large and widespread group of enzymes frequently found in bacterial pathogens, bacterial human gut symbionts, and a variety of fungi[29]. Despite the low amino acid sequence identity between GH125 and GH15 family members, shared structural aspects of their catalytic centers suggest that family GH125 belongs to the GH-L glycoside

**Fig. 2 Biochemical characterization of *M. hassiacum* MmpH. A** MMP biosynthetic cluster and schematic representation of the hydrolytic activity of MmpH using MMP as substrate. **B** TLC analysis of a time course of MmpH activity; C, control reaction without enzyme. **C** TLC analysis of eluted fractions after purification of MmpH oligomannoside products; Lane R, MmpH reaction mixture using MMP as substrate. The purest oligomannoside products (**a-d**) eluted are indicated by black boxes. **D** Positive ion mode ESI-TOF analysis of purified oligomannoside products. Product ion nomenclature follows that proposed by Domon and Costello[82] and the fragmentation pathways of Y-type glycosidic cleavages are indicated. The representation of glycans follows the guidelines of Symbol Nomenclature for Glycans[80]. A filled green circle represents a mannose residue. **E** Temperature profile of MmpH activity. **F** pH dependence of MmpH activity in the presence of MES (triangle) and BTP (circle) buffers. **G** Michaelis–Menten curve for the enzymatic activity of MmpH at 37 °C. **H** Michaelis-Menten curve for the enzymatic activity of MmpH at 45 °C. **I** Relative activity of MeT1 using natural MmpH products (oligomannosides **a-d**) and synthetic sMetMan₂ as substrate. **J** Enzymatic activity of MmpH variants Asp47Ala (D47A), Asp50Ala (D50A) and Glu256Ala (E256A) using MMP as substrate. **K** TLC analysis of the activity (30 min) of MmpH variants using MMP as substrate; C, control reaction without enzyme; Products **a** and **b** are in black boxes. Error bars represent standard deviation of three replicates.

**Table 1 Kinetic properties of MmpH, ManT and ManT Glu313Ala variant.**

| T (°C) | Enzyme | Substrate | Saturating concentration | $K_m$ (mM) | $V_{max}$ (μmol/min/mg) | $k_{cat}$ (s$^{-1}$) | $k_{cat}/K_m$ (mM$^{-1}$.s$^{-1}$) |
|---|---|---|---|---|---|---|---|
| 37 | MmpH | MMP | | 0.28 ± 0.05 | 0.76 ± 0.03 | 0.011 ± 0.001 | 0.039 ± 0.007 |
| 45 | | | | 0.36 ± 0.06 | 1.42 ± 0.05 | 0.021 ± 0.001 | 0.058 ± 0.001 |
| | ManT WT | sMetMan₄ | 2.5 mM GDP-Man | 0.13 ± 0.03 | 1.50 ± 0.04 | 0.047 ± 0.001 | 0.362 ± 0.068 |
| | | GDP-Man | 1.5 mM sMetMan₄ | 0.28 ± 0.05 | 1.55 ± 0.08 | 0.048 ± 0.002 | 0.171 ± 0.033 |
| 37 | | sMan₄ | 5 mM GDP-Man | 2.78 ± 0.45 | 0.80 ± 0.05 | 0.025 ± 0.001 | 0.009 ± 0.002 |
| | | GDP-Man | 2.5 mM sMan₄ | 0.49 ± 0.08 | 0.57 ± 0.02 | 0.017 ± 0.001 | 0.035 ± 0.006 |
| | ManT Glu313Ala variant | sMetMan₄ | 2.5 mM GDP-Man | 1.18 ± 0.30 | 0.21 ± 0.02 | 0.006 ± 0.001 | 0.005 ± 0.001 |
| | | GDP-Man | 3.5 mM sMetMan₄ | 1.23 ± 0.30 | 0.31 ± 0.03 | 0.009 ± 0.001 | 0.007 ± 0.002 |

hydrolase clan. In particular, there is remarkable conservation of the molecular determinants of sugar recognition at the (-1) subsite. Close inspection of this subsite and comparison with G1d and CpGH125 revealed only partial structural conservation in MmpH (Fig. 3E, F). The catalytic base (Glu628 in G1d and Glu393 in CpGH125) seems to be conserved in MmpH (Glu262), as is the network of solvent molecules coordinated to these residues and that participate in the inverting reaction mechanism. In agreement, replacing Glu262 of MmpH with an alanine residue completely abolished its activity (Fig. 2J, K). On the other hand, two residues of MmpH are relatively close to the position of the catalytic acid in G1d (Glu430) and CpGH125 (Asp220): Asp47, located in the loop connecting helices αA and αB, and Asp50, which could also play a role in sugar recognition (see below). The drastic reduction of activity observed for both Asp47Ala and Asp50Ala MmpH variants is compatible with one of these residues being the catalytic acid (Fig. 2J, K). Given the spatial proximity of these two aspartate residues in the active center of MmpH and their extensive interactions, replacement of either of them could lead to subtle local rearrangements, destabilizing the enzyme and impacting its activity or allowing for partial functional compensation by relocation of the neighbor side chain, in line with the much reduced activity level observed for both variants. The position occupied by the side chain of Trp315 in MmpH corresponds to those of Trp330 and Trp62 in G1d and CpGH125, respectively, although with a rotation (up to approximately 90°) of the plane of the indole ring. In G1d and CpGH125, the side chain of the tryptophan stacks with and stabilizes the sugar rings positioned at the (−1) subsite, a plausible role also for Trp315 in MmpH, provided that the (−1) methylmannose unit of MmpH is adequately oriented (i.e., perpendicular to that of G1d and CpGH125 ligands) for processing or there is a reorientation of the side chain of Trp315 upon binding of MMP to MmpH. The residues that interact with the O3 and O4 hydroxyl groups of the non-reducing sugar ring at subsite (−1)—Arg332 and Asp333 in G1d and Arg64 and Asp65 in CpGH125—are absent in MmpH. Residues Arg567 (in G1d) and Asn302 (in CpGH125) make polar contacts with the equatorial O2 hydroxyl of (−1) glucose and mannose sugar rings, respectively. Interestingly, in MmpH the volume corresponding to those residues is part of its large depression and could accommodate an additional sugar moiety, in line with its endo-mannosidase activity and in contrast with the exo-mannosidase

properties of GH125 family members. Indeed, in both MmpH protomers there is additional electron density at this region that was interpreted as bound glycerol molecules (one per MmpH molecule). This bound glycerol likely mimics half of the sugar ring binding to this potential subsite and consequently the interactions observed could partially define the local molecular determinants of substrate selection (Fig. 3G). One of the hydroxyl groups of the glycerol molecule establishes a direct polar interaction with the side chain of Asn101, while water-mediated contacts connect the side chains of Asp47, Asp50, Glu262 and Glu265 with the other hydroxyl groups of the ligand. Moreover, the side chains of Trp49 and Phe102 on one side and those of Thr163, Met215 and Tyr219 on the other, define a hydrophobic environment that could be important to accommodate an additional mannose unit at this subsite. All the residues defining this binding site are highly conserved, not only in mycobacterial MmpH homologues but also in those from *Corynebacterium*, *Rhodococcus*, *Streptomyces*, and *Nocardia* (Fig. 3C, Fig. S3B, Table S5, and Fig. S7).

**ManT is a highly specific α-(1→4)-mannosyltransferase.** A gene (*manT*) in the *M. hassiacum* MMP cluster encodes a putative glycosyltransferase with high amino acid sequence identity to those of related actinobacteria, namely >75% to other NTM, 53% to that from *Streptomyces griseus* in which a related MMP was identified, and 69% to the homologue from *Nocardia otitidisca-viarum*, a bacterium not known to synthesize MMP[9]. The enzyme (ManT) displayed low-to-moderate amino acid sequence identity (<35%) to other mannosyltransferases involved in the synthesis of mycobacterial mannoglycans, namely PIM, LM and LAM, none of which containing α-(1→4) linked mannoses (Fig. S8 and Table S6). Moreover, ManT displayed low amino acid sequence identity (28–33%) with mycobacterial α-(1→4)-glycosyl-transferases (Rv3032 and Rv1212c) involved in the synthesis of the α-(1→4)-linked glucans MGLP or glycogen.

The products of MMPs hydrolysis by MmpH were suitable substrates for ManT (Fig. 4A). The substrate specificity of ManT was assessed in vitro using mono- and dimannosides (methylated at positions 1-OH and/or 3-OH) and four synthetic mannosides (sMan₄, sMan₃, sMetMan₄ and sMetMan₃), differing in length and in the methylation (Met) status of position 3-OH (Fig. 4B, C

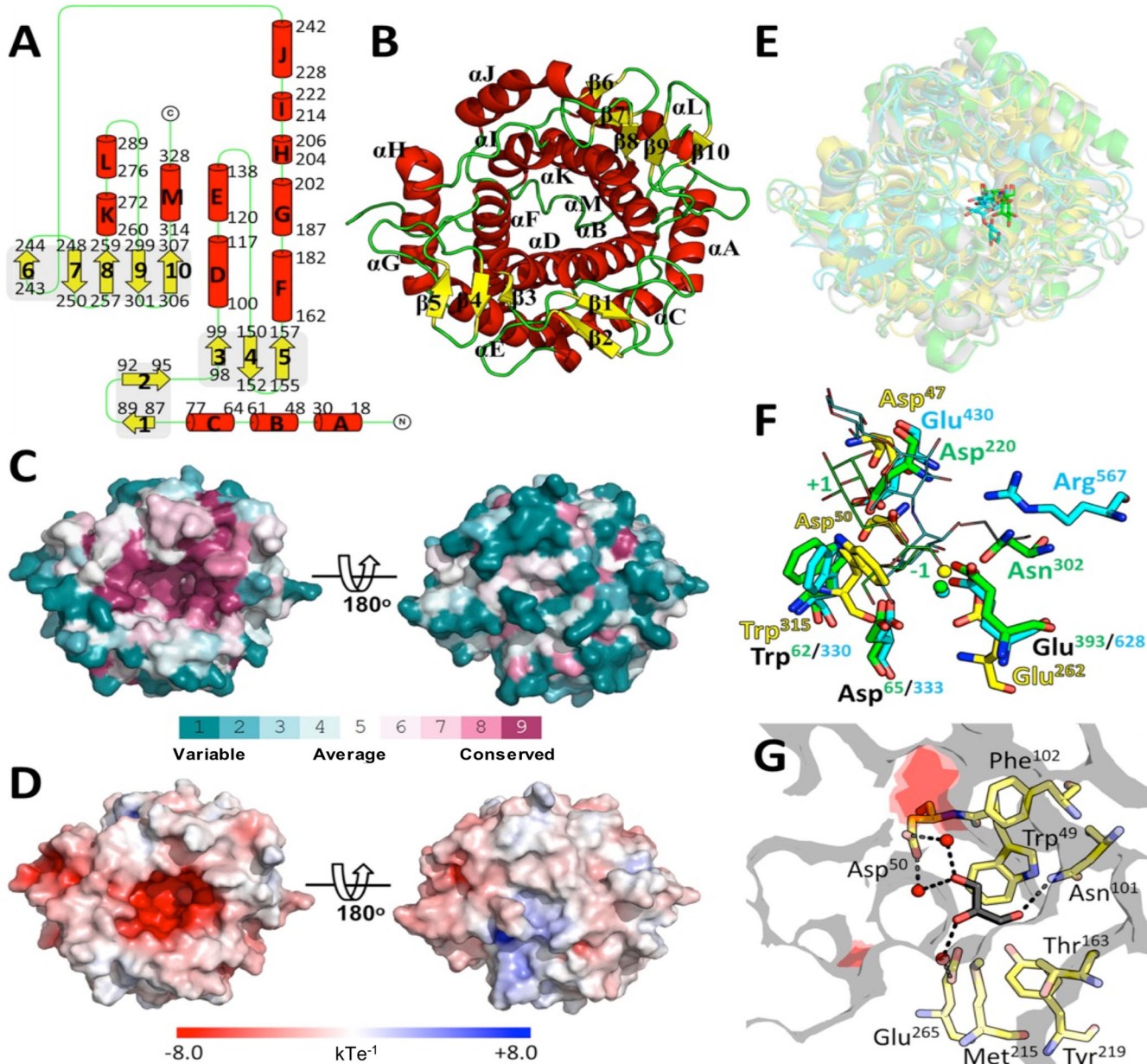

**Fig. 3 Overall structure of *M. hassiacum* MmpH and structural comparison. A** Topology diagram of MmpH with α-helices (A–M) and β-strands (1–10) represented as red cylinders and yellow arrows, respectively. The number of the first and last residue of each secondary structure element is indicated. **B** Cartoon representation of the MmpH monomer with secondary structure elements colored as in panel (**A**). **C** Solid surface representation of MmpH colored according to ConSurf[83] residue conservation score, calculated from the amino acid sequence alignment of 150 MmpH homologues (fig. S7). The color-code gradient (from teal for variable positions to purple for highly conserved ones) is shown at the bottom of the panel. The left pose is oriented as in panel **B** (top view). The right pose results from a 180° rotation of the left pose around *x* (bottom view). **D** Solid surface representation of MmpH colored according to its electrostatic surface potential (calculated with APBS[84], as implemented in PyMOL (Schrödinger)), contoured from −8 (red) to 8 kT/e (blue). The two views of the MmpH monomer are those depicted in panel (**C**). **E** Superposition of MmpH (yellow) in the same orientation as in panel **B** with GH15 family member-G1d from *Arthrobacter globiformis* (cyan; PDB entry 1ULV[28]), CpGH125 from *Clostridium perfringens* (green; PDB entry 3QT9[29]) and SpGH125 from *Streptococcus pneumoniae* (grey; PDB entry 3QRY[29]). All proteins are shown in cartoon representation and the ligands acarbose, α-1,6-linked 1-thio-α-mannobiose and 1-deoxymannojirimycin (dMNJ) as stick models with carbon atoms colored as the corresponding cartoon, nitrogen atoms blue, oxygen red and sulfur yellow. **F** Close-up view of the superposed (-1) sugar binding subsites of G1d, CpGH125 and MmpH. Most residues involved in sugar recognition, as well as the catalytic acid and base are shown as stick models (color code as in panel **E**). Water molecules important for the inverting reaction mechanism are shown as spheres and colored as the carbon atoms of the corresponding protein. Sugar-binding subsites (−1) and (+1) of α-1,6-linked 1-thio-α-mannobiose-binding CpGH125 are indicated for clarity. **G** Close-up of the adjacent subsite of MmpH, represented as a transparent grey surface with the putative catalytic acid (Asp47) and base (Glu262) colored red. Residues establishing contacts with the bound glycerol molecule are represented as stick models and colored as in panel (**F**). Dashed black lines and red spheres represent hydrogen bonds and water molecules, respectively.

and Fig. S9A). ManT was active toward the four MmpH products (**a-d**), preferring the natural pentamannoside **b** and the synthetic tetramannoside sMetMan$_4$ over the natural tetramannoside **a** (Fig. 4A, C). Indeed, although both substrates possess four methylated mannose units, ManT displayed 20% lower activity with tetramannoside **a** than with the synthetic sMetMan$_4$ (Fig. 4C). Further, ManT displayed low activity with the natural octamannoside **d**, comparable to that observed with the synthetic unmethylated substrates sMan$_4$ and sMan$_3$ (Fig. 4C). Although it was not quantified due to the scarcity of this substrate, ManT activity with heptamannoside **c** seems to be comparable to that observed with octamannoside **d** (Fig. 4A).

In agreement with earlier studies, *M. hassiacum* ManT could use synthetic trimannosides and tetramannosides as substrates, but not mono- and disaccharides[21,23], and was significantly more active with the methylated substrates, preferring the tetramanno-sides over the trimannosides (Fig. 4C). Moreover, ManT was specific for mannosyl acceptors as it did not use maltoligosaccharides and was specific for the α configuration of the glycosidic bond, being inactive with all of the β-(1→4)-mannosides tested.

Analysis of ManT reactions using synthetic tri- and tetra-mannosides as substrates revealed the formation of two products from sMetMan$_3$ and sMan$_4$, while three products were formed in the presence of sMan$_3$ and sMetMan$_4$ (Fig. 4D and Fig. S9A). Product identification corroborated the sequential addition of mannose units to the acceptor oligomannoside substrates by ManT (Fig. S9B, C). Products sP1, sP2 and sP3 represent the starting tetramannoside (sMetMan$_4$) plus one (sP1), two (sP2) or three (sP3) additional mannoses, i.e., newly formed penta-, hexa- and heptamannoside (Fig. 4E).

**Biochemical and kinetic properties of ManT**. Purification of bioactive ManT (approximately 2 mg pure enzyme/L culture) required the addition of a detergent (Triton X-100) to all buffers. ManT was active between pH 6.5 and 10, peaking at pH 8.5 (Fig. 4F) and between 20 and 55 °C with maximum activity at 40 °C (Fig. 4G). However, all kinetic parameters were determined at 37 °C, a temperature relevant for human mycobacterial infections and at which ManT displayed 95% of its maximal activity. Similar to other GT-B fold enzymes that require divalent cations for full activity, although without evidence of a bound metal ion associated with catalysis[30], 7.5 mM MgCl$_2$ significantly enhanced (approx. five-fold) the activity of ManT that was only slightly increased by NaCl (Fig. 4H, I). Unsurprisingly, the activity of ManT was enhanced by Triton X-100 (Fig. 4J), indispensable for stabilizing the enzyme in solution and that likely mimics the hydrophobic environment provided by the membrane fraction where the activity was initially detected[21,23]. The kinetic characterization of ManT, with varying concentrations of GDP-Man and with both the unmethylated and 3-O-methylated tetra-mannosides (Fig. 4K–N), revealed a marked preference for the methylated tetramannoside (twenty-two times lower apparent $K_m$ and higher $V_{max}$ for sMetMan$_4$ than for sMan$_4$), which results in a 40-fold increase in catalytic efficiency when using sMetMan$_4$ over sMan$_4$ (Table 1).

**Structure of ManT**. The structure of recombinant ManT from *M. hassiacum* was determined at 2.75 Å resolution from monoclinic (space group C2) crystals containing two molecules in the asymmetric unit. ManT belongs to the GT4 family of retaining glycosyltransferases and displays the typical GT-B fold[31], consisting of two Rossmann domains connected by a linker. Most of the C-terminal domain could be modeled in both molecules, while the N-terminal domain was almost complete only in molecule A (hereafter, the structure of molecule A is described,

unless noted). Accordingly, the N- (Met1-Val209 and Trp390-Cys411) and C-terminal (Asn227-Phe388) domains of ManT display a core formed by parallel β-strands (β1-13) alternating with connecting α-helices (αA-O) and form a deep crevice at the inter-domain interface (Fig. 5A).

The closest structural homologues of ManT, as identified using the Dali protein structure comparison server[27], are the glycosyl-transferase MshA from *Corynebacterium glutamicum* (PDB entry 3C4Q[32], Z score = 37.2, r.m.s.d. = 2.8 Å, 24% amino acid sequence identity) and the GDP-Man dependent phosphatidyli-nositol mannosyltransferase PimA from *Mycobacterium smegmatis* (PDB entry 2GEJ[33], Z score = 32.7, r.m.s.d. = 2.9 Å, 25% amino acid sequence identity). Despite low amino acid sequence conservation (Fig. S8B and Table S6), the three enzymes belong to the GT4 family and display a GT-B fold (Fig. 5B). They display particularly striking structural conservation in the nucleotide-binding C-terminal domain[30] that allowed the identification of residues potentially involved in GDP-Man recognition (Fig. 5B–D). The conservation of the nucleotide-binding cleft, with its slightly positive electrostatic surface potential, becomes clear when mapping residue conservation on the ManT surface (Fig. 5E, F and Fig. S10). Thus, most of the molecular determinants for binding of GDP-Man and UDP-GlcNAc to PimA and MshA, respectively, are structurally conserved in ManT (Fig. 5C, D). In PimA, the guanidyl N2 nitrogen of GDP-Man makes polar contacts with the carbonyl group of Val251 and the side chain of Asp253, interactions that could be maintained in ManT with Leu291 and Asp293. In contrast, nucleotide recognition in MshA includes the establishment of hydrogen bonds between the uracil base of UDP and the main-chain of Arg294, structurally equivalent to Val251 and Leu291 of PimA and ManT, respectively. A conserved glutamate residue has been reported in GT-B fold members, which interacts with the ribose moiety of the sugar-donor[34]. In PimA and MshA, this role is played by Glu282 or Glu324, respectively, that establish a bidentate contact with oxygens O2' and O3' of GDP-Man (in PimA) or of UDP-GlcNAc (in MshA). This signature residue is also conserved in ManT (Glu321) and its critical role in catalysis is underscored by the complete lack of activity of the Glu321Ala ManT variant (Fig. 5G). The pyrophosphate oxygens of the sugar donors are stabilized by a network of hydrogen bonds, in both PimA (with Gly16, Arg196 and Lys202) and MshA (with Gly23, Arg231 and Lys236). Two of these residues are structurally conserved in ManT: Gly15, in the characteristic GT-B fold glycine-rich loop motif[34] and Lys240, located in the partially disordered β8-αH loop and occupying a position similar to that of Lys202 in PimA and Lys236 in MshA. Presumably due to the absence of a ligand, this region of ManT displays high mobility and poor electron density maps that impaired modeling part of the loop (Ser234-Pro238). Upon GDP-Man binding, it is conceivable that a rearrangement of the β8-αH loop reorients the side-chain of Lys240 to the catalytic interface, anchoring the distal phosphate of the sugar donor, as suggested by the inactivity of the ManT Lys240Ala variant (Fig. 5G, H).

In the PimA-GDP-Man complex the mannosyl moiety is stabilized by several polar contacts between the hydroxyl groups of the sugar ring and main-chain protein atoms of Ser275, Phe276 and Ile278 and the carboxylate group of Glu274 (Fig. 5D). All of these residues are structurally conserved in ManT—Gly314, Phe315, Leu317 and Glu313 (Fig. 5D)—and located within the flexible β11-αK loop that seems to adopt different conformations to accommodate the mannosyl moiety of the sugar-donor, in line with the two poses observed for ManT molecules A and B. As a result, in molecule B, Glu313 is close to the position occupied by Glu274 in PimA (Glu316 in MshA). It is worth noting that this glutamate residue is part of the signature motif of the

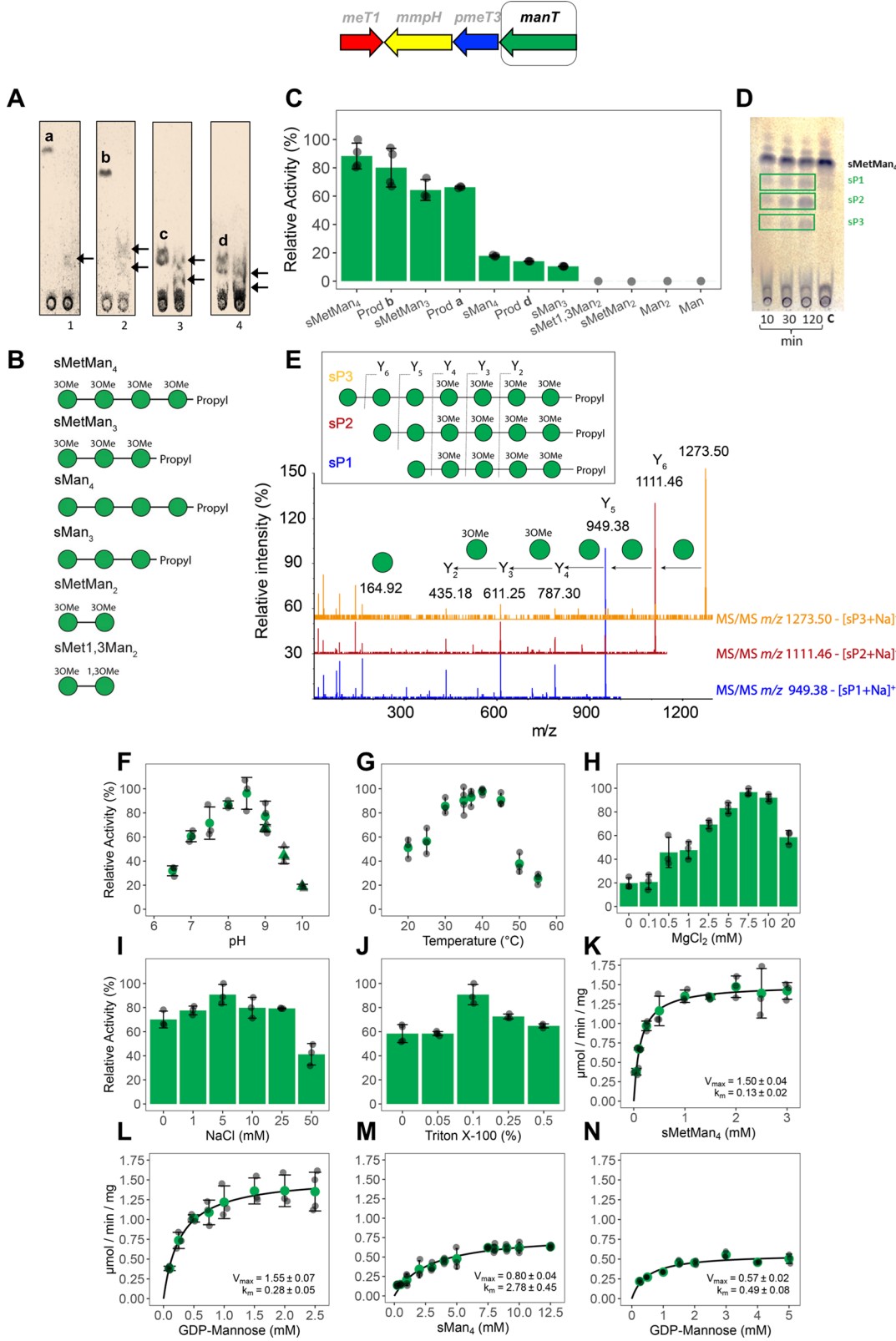

GPGTF/GT-B superfamily and is thought to be important for catalysis in homologous GT-B glycosyltransferases[33,34]. In agreement, the Glu313Ala ManT variant displays drastically reduced catalytic activity (Fig. 5G and Table 1) with an apparent $K_m$ for GDP-Man approximately 4-fold higher than the wild-type enzyme, which agrees with the contribution of Glu313 to the recognition of the sugar-donor and, consequently, to catalysis (Fig. 5I, J and Table 1).

The N-terminal domains of enzymes of the GT-B family have evolved to accommodate very different acceptors, frequently originating structural differences, particularly in the loops and helices pointing towards the active site interface[30]. However,

**Fig. 4 Biochemical characterization of *M. hassiacum* ManT. A** TLC analysis of ManT activity using the natural MmpH oligomannoside products as substrates. Letters **a** to **d** indicate products of MmpH activity used in control reactions without ManT. Lanes 1-4, ManT reactions with tetramannoside **a**, pentamannoside **b**, heptamannoside **c** and octamannoside **d**. Products of ManT activity are indicated by arrows. **B** Schematic representation of chemically synthesized oligomannosides (see Supplementary Methods). Green circles represent mannose units. **C** Relative activity of ManT in the presence of natural oligomanosides (MmpH products) and synthetic acceptor substrates: Man (mannose); $Man_2$ (α-mannobiose); $sMetMan_2$[24], $sMet1,3Man2$[24], $sMan_3$, $sMan_4$, $sMetMan_3$ and $sMetMan_4$ (see Supplementary Methods). **D** TLC analysis of a time course of ManT activity using the synthetic tetramannoside $sMetMan_4$ as substrate: C: control reaction without enzyme; sP1, sP2 and sP3 are products of ManT activity using synthetic substrates. **E** MS/MS analysis of ManT products sP1 to sP3 (schematic representation in inset). Product ion nomenclature follows that proposed by Domon and Costello[82] and the fragmentation pathways of Y- type glycosidic cleavages are shown. The representation of glycans follows the guidelines of Symbol Nomenclature for Glycans[80]. A filled green circle represents a mannose residue. **F** pH dependence of ManT activity examined in BTP (circles) and CAPSO (triangles) buffers. **G** Temperature profile of ManT activity. **H** ManT activity in the presence of increasing concentrations of $MgCl_2$. **I** Effect of NaCl concentration on the activity of ManT. **J** Effect of Triton X-100 concentration on the activity of ManT. Biochemical properties were determined in the presence of the tetramannoside acceptor $sMetMan_4$. **K, L** Michaelis–Menten curve for the activity of ManT as a function of substrates $sMetMan_4$ and GDP-Man, respectively. **M, N** Michaelis–Menten curve for the activity of ManT as a function of unmethylated $sMan_4$ and GDP-Man, respectively. Kinetic parameters were examined at 37 °C. Error bars represent standard deviation of three replicates.

there is a largely conserved histidine residue in the N-terminal domain of GT-B proteins, which is also part of the GPGTF signature motif and involved in catalysis[34,35]. In ManT, His142 is not only conserved but is structurally equivalent to His118 of PimA—essential for mannosylation[33]—and to His133 of MshA, hinting at a conserved functional role for this amino acid.

**MmpH is essential for MMP biogenesis**. To confirm the function of MmpH and the proposed MMP recycling mechanism, we generated an unmarked *mmpH M. smegmatis* mutant (*MsmegΔmmpH*), unable to produce MMP, linking MmpH function to MMP biogenesis (Fig. 6A–C and Fig. S11A). Under optimal growth conditions, absence of MMP did not impair growth of the mutant, or its MGLP production pattern (Fig. S11). Due to the proposed involvement of MGLP in mycobacterial resistance to thermal stress[10] and to the apparent functional redundancy of the two polysaccharides in modulating fatty acid metabolism[36], we sought to examine a possible role of MMP during mycobacterial growth both at sub- and supraoptimal temperatures, known to influence the lipidic composition of the cell envelope[37]. The growth rate of the mutant at 37 and 45 °C was comparable (Fig. 6D). However, the growth rate of the mutant at 15 °C was lower than that of the WT (18% higher average generation time (GT)) (Fig. 6C, D and Fig. S11B). Despite the growth rate discrepancy, no obvious differences could be detected on the levels of intracellular MGLP (Fig. S11A).

**Discussion**
The polymethylated polysaccharides MGLP and MMP, initially identified in *M. smegmatis* and *M. phlei*[8,9], were proposed to modulate fatty acid metabolism and the assembly of cell envelope lipids[36,38]. While MGLP biosynthesis appears to occur in all mycobacteria, MMP is absent from *M. tuberculosis* and has only been isolated from rapidly growing NTM[9]. The escalation in the number of infections by NTM, the growing number of people at increased risk, and the poor therapeutic options available to treat NTM disease, prompted a detailed study of MMP not only to disclose its real distribution in mycobacteria, but also its biosynthetic mechanism and producing enzymes, as well as its role in mycobacterial physiology and in the adaptation to environmental stress. We previously identified the gene cluster responsible for MMP biosynthesis and characterized a highly specific 1-*O*-methyltransferase (MeT1), defining a key step in the biogenesis of this unique polysaccharide[24]. In this work, we unveil and characterize the functions of a unique and highly specific MMP hydrolase (MmpH), and of a rare mannosyltransferase (ManT) that catalyzes the formation of α-(1→4) bonds between the mannose units in the MMP chain.

Recombinant *M. hassiacum* MmpH cleaved with precision the main backbone of $MMP_{11-14}$ purified from *M. smegmatis* into distinct oligomannosides. The asymmetric cleavage generated 1-*O*-methyl-blocked 3-*O*-methylated tetra- and pentamannosides representing the MMP reducing end, and 3-*O*-methylated hepta- and octamannosides matching the non-reducing terminal of MMP. Therefore, MmpH is a genuine α-(1→4)-endomannosidase with the ability to hydrolyze α-(1→4) linked 3-*O*-methyl-mannose linear polymers.

Early experiments using *M. smegmatis* cell extracts showed that ManT used only acceptors longer than a tetramannoside ($MetMan_4$)[21], coincidentally also the smallest MMP precursor isolated[25] and thus suggesting that the initial steps in MMP biosynthesis were likely independent of ManT activity. Subsequent studies also in cell extracts identified substrates longer than trimannosides ($Man_3$ or $MetMan_3$) as suitable acceptor substrates for ManT[23]. Our results show that ManT can transfer mannose units to a wide range of α-(1→4) oligomannosides longer than three mannoses, including all hydrolytic products of MmpH, making it the only known mannosyltransferase able to catalyze formation of α-(1→4) bonds between mannose residues. Unlike previously suspected, we also found that ManT was not able to elongate methylated mono or dimannosides, including the 1,3-dimethylmannoside obtained by selective methylation by MeT1[24]. Maximal ManT activity was observed with the natural MmpH pentamannoside product (or with the synthetic $sMetMan_4$) as acceptor, followed by the natural tetramannoside (or the synthetic $sMetMan_3$), with longer natural substrates and the unmethylated synthetic acceptors ($sMan_4$ and $sMan_3$) being much disfavored. The difference in the number of mannose units from the preferred natural and synthetic substrates could be explained by the presence of a propyl group at the reducing end of the latter, possibly mimicking the presence of an additional mannose residue within the active site pocket of the enzyme. Further, we confirmed that ManT could add up to three mannose units sequentially and independently of alternating methylations at position 3-OH, in agreement with a previously proposed mechanism[23]. However, since ManT products longer than hexa- or heptamannosides could not be detected in vitro, the alternating mannosylation/methylation mechanism[21,22] cannot be discarded as a pre-requisite for optimal synthesis of the longer mature MMP. It is also possible that the full-length extension of MMP may require addition of preassembled methylmannose oligomers.

The oligomannoside products of MMP hydrolysis by MmpH were specific substrates both for ManT and MeT1, unequivocally linking the genes in the cluster to MMP biogenesis. While ManT could use all four MMP oligomannosides fragments as acceptors, it preferred the 1-*O*-methylated pentamannoside. Unsurprisingly,

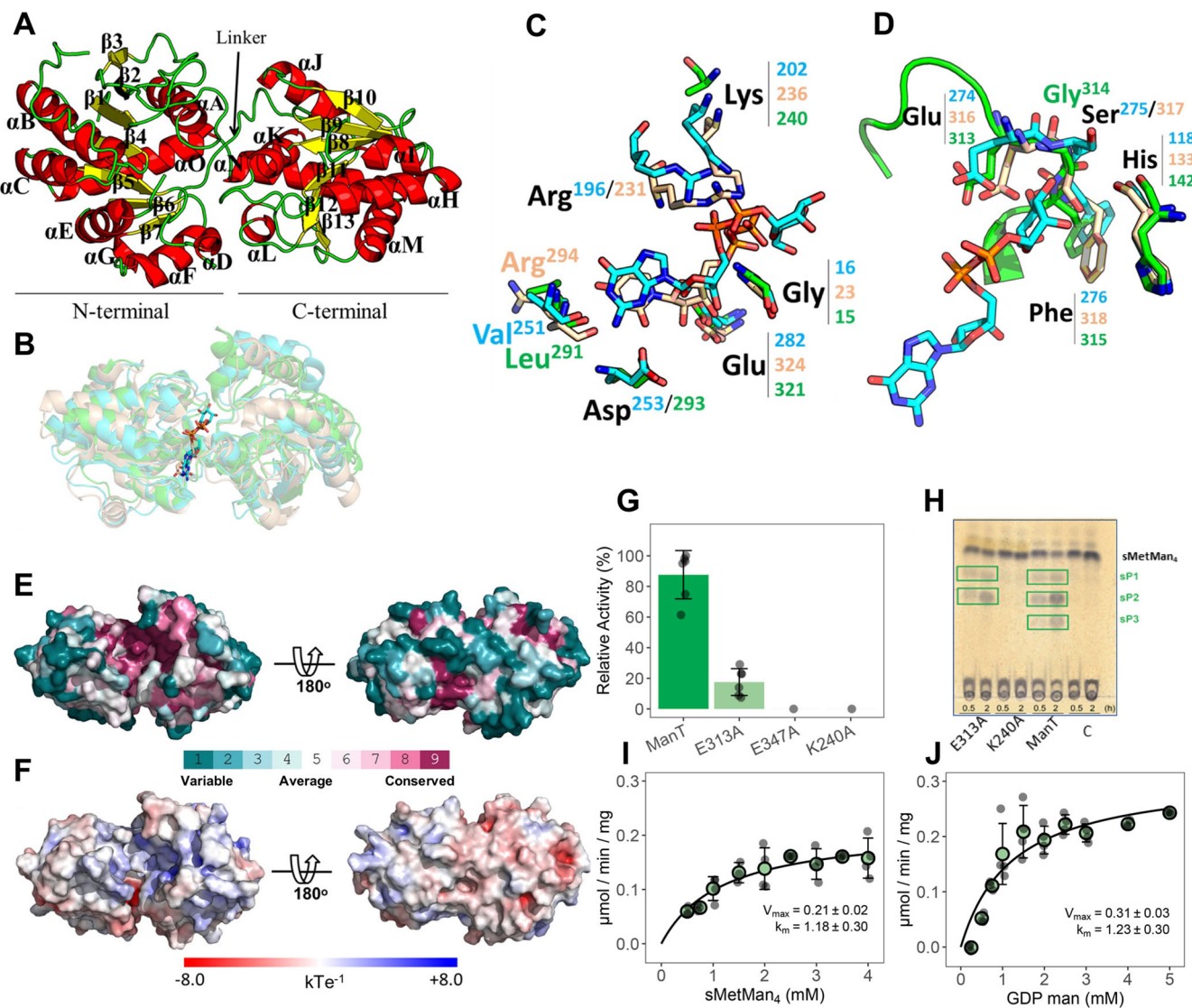

**Fig. 5 Structural characterization of *M. hassiacum* ManT. A** Cartoon representation of the crystallographic structure of ManT with secondary structure elements colored yellow (β-strands β1–β13) and red (α-helices αA–αO). The N- and C-terminal domains of the protein and the inter-domain linker are also labeled. **B** Superposition of the three-dimensional structure of ManT with those of structural homologues. The crystal structure of ManT (green) was superposed with those of GT-4 family members PimA from *M. smegmatis* (cyan; PDB entry 2GEJ[33]), and MshA from *Corynebacterium glutamicum* (tan; PDB entry 3C4Q[32]). All proteins are represented as transparent cartoons and bound GDP-Man and UDP are represented as stick models with carbon atoms colored as the corresponding cartoon, nitrogen atoms blue and oxygen red. **C, D** Close-up views of the sugar-donor binding sites of PimA, MshA and ManT. Residues directly involved in the recognition of the nucleotide (**C**) and of the mannosyl moiety of GDP-Man (**D**) are shown as stick models and colored as in (**B**). The β11-αK loop of ManT, in the conformation observed in molecule B is also represented. **E** Solid surface representation of ManT colored according to ConSurf[83] residue conservation score, calculated from the amino acid sequence alignment of 149 ManT homologues (fig. S10). The color-code gradient (from teal for variable positions to purple for highly conserved ones) is shown at the bottom of the panel. The left pose is oriented as in **B** (top view). The right pose results from a 180° rotation of the left pose around *x* (bottom view). **F** Solid surface representation of ManT colored according to its electrostatic surface potential (calculated with APBS[84], as implemented in PyMOL (Schrödinger)), contoured from −8 (red) to 8 kT/e (blue). The two views of the ManT monomer are those depicted in panel (**E**). **G** Relative activity of ManT variants Glu313Ala (E313A), Glu321Ala (E321A) and Lys240Ala (K240A) using sMetMan₄ and GDP-Man as substrates. **H** TLC analysis of the activity of ManT variants (30 min and 2 h) using sMetMan₄ as substrate; C, control reaction without enzyme. ManT products sP1 to sP3 are in green boxes. **I, J** Michaelis–Menten curve for ManT Glu313Ala variant as a function of substrates sMetMan₄ and GDP-Man, respectively. Kinetic parameters were examined at 37 °C. Error bars represent standard deviation of three replicates.

MeT1 could only methylate the oligomannoside fragments with a free 1-OH group. These collective observations suggest that MMP biogenesis relies on a partially conservative mechanism that depends on pre-existing mature MMP and on the concerted action of a MMP endomannosidase (MmpH), a unique mannosyltransferase (ManT) and highly specific methyltransferases (Fig. 7). In this elegant mechanism, MmpH serves as the recycling enzyme that hydrolyses mature MMP into defined-size smaller

oligomannosides that are, in turn, substrates for ManT and MeT1 activities for further processing into new daughter MMP chains. It is very likely that pMeT3 encoded in the MMP gene cluster is the missing 3-O-methyltransferase for 3-O-methylation of the mannose units added by ManT and required for full maturation of MMP (Fig. 7)[24].

The hypothesis of a self-sustained and partially conservative MMP biogenesis mechanism, driven by the hydrolytic activity of

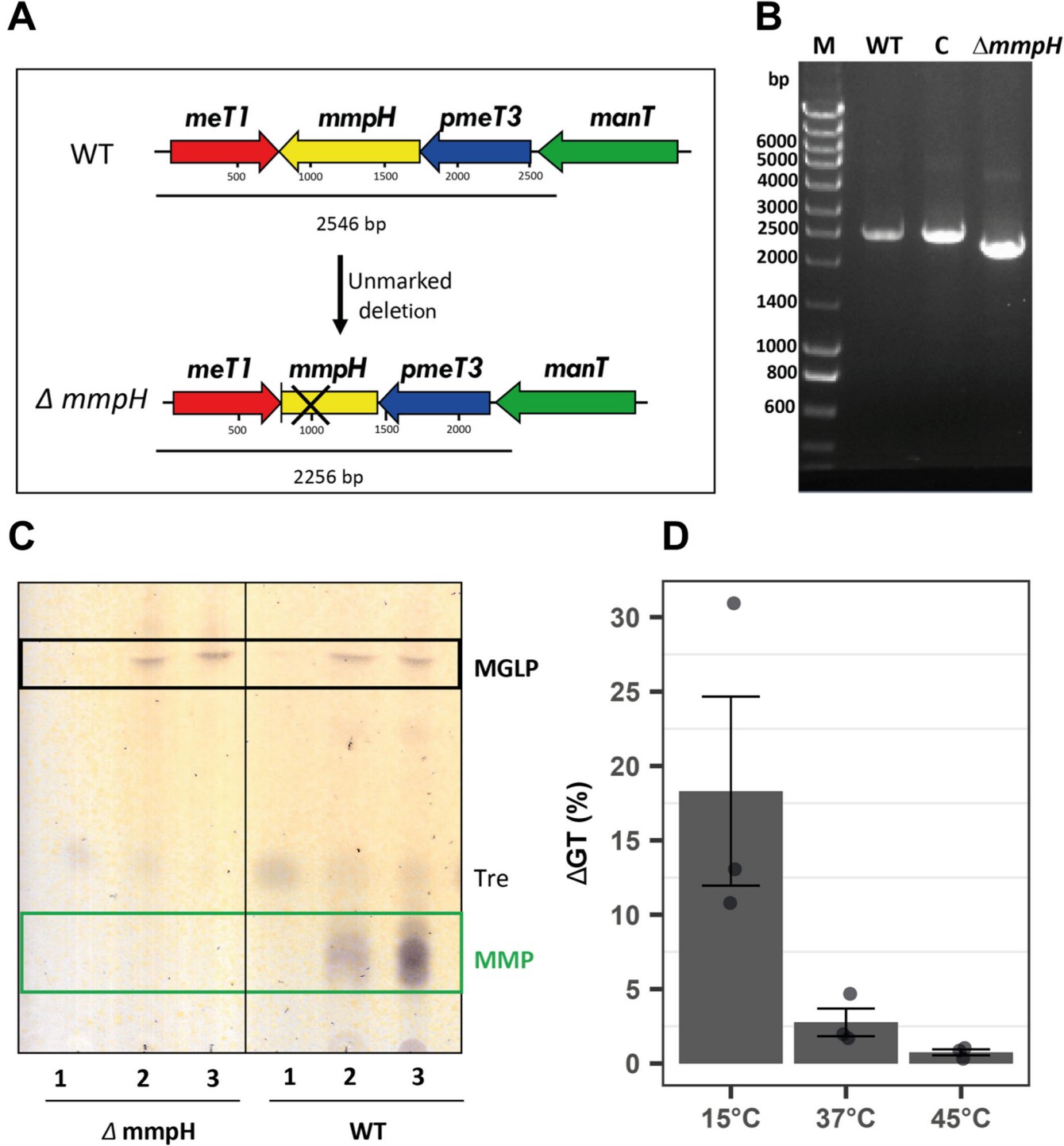

**Fig. 6 Effects of the inactivation of *mmpH* on the growth of *M. smegmatis*. A** Genetic organization of *mmpH* and neighboring genes in *M. smegmatis* and in the unmarked *mmpH*-negative mutant. The deletion site on *mmpH* is marked by a cross. **B** PCR confirmation of *M. smegmatis* p2NIL-Δ*mmpH*-sel transformants. M, molecular weight marker; WT, amplification from *M. smegmatis* DNA; C, Msmeg:p2NIL-Δ*mmpH*-sel negative transformant; Δ*mmpH*, successful transformant with evident loss of a 290 bp fragment from the *mmpH* gene. **C** TLC analysis of PMPS purified from WT and Δ*mmpH M. smegmatis*. Tre, trehalose. Lanes 1, 2, 3 elution with 40, 60 and 80% (v/v) methanol. **D** Relative difference between the generation time (GT) of WT *M. smegmatis* and the Δ*mmpH* mutant (Fig. S11B) grown at the indicated temperatures in GBM. Error bars represent SEM of three replicates.

MmpH, is also supported by the phenotype of the MmpH-negative mutant strain generated in this study. MMP had previously only been isolated from RGM, leading to the speculation that it could contribute to their faster growth. However, we show that the distribution of the MMP gene cluster is not restricted to RGM and that MMP is indeed synthetized in some SGM

including the important opportunistic pathogen *M. avium*. Further, compared to the wild-type organism, the MMP-negative mutant displayed a higher generation time (lower growth rate) at 15 °C but not at optimal and supra-optimal temperatures. This overruled the hypothesis of a direct role of MMP in the higher growth rate of RGM but possibly implicated this polysaccharide

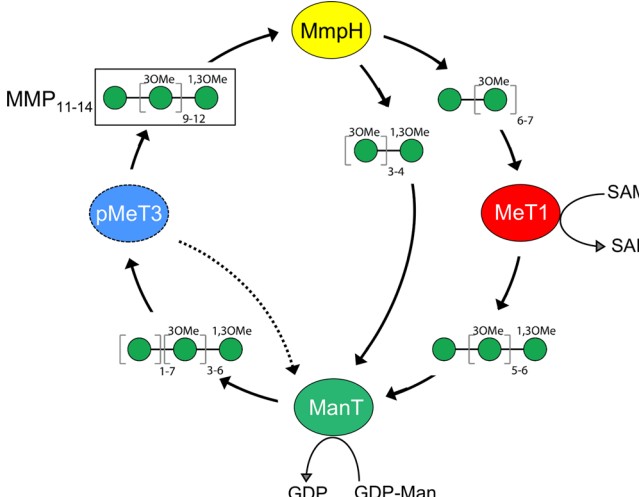

**Fig. 7 Proposed self-recycling mechanism for partially conservative biogenesis of mycobacterial MMP.** This mechanism relies on the hydrolytic activity of MmpH toward MMP, followed by the synthesis of new daughter MMP molecules occurring by the coordinated action of ManT, MeT1 and pMeT3. A dashed arrow indicates putative alternating ManT and pMeT3 activities proposed to be required for full assembly of mature MMP[21,22]. MeT1, MMP 1-O-methyltransferase; MmpH, MMP hydrolase (α-endomannosidase); pMeT3, putative MMP 3-O-methyltransferase; ManT, MMP α-(1→4)-mannosyltransferase; SAM, S-adenosyl methionine; SAH, S-adenosyl homocysteine; GDP-Man, Guanosine 5'-diphospho-α-D-mannose; GDP, Guanosine 5'-diphosphate.

in mycobacterial adaptation to cold stress, unlike MGLP, which was found to be associated to adaptation to heat stress[10]. These findings strengthen the hypothesis that the existence of two similar and apparently redundant polymethylated polysaccharides is probably due to them having specialized functions[39]. In summary, we reveal the identity and the biochemical and structural properties of a rare α-(1→4)-endomannosidase with an essential role in MMP self-recycling, and also of a unique α-(1→4)-mannosyltransferase for MMP elongation, both crucial elements of a novel recycling and partially conservative replication mechanism for MMP biogenesis in NTM. We propose that this probably energetically favorable recycling mechanism may favor MMP synthesis and availability during mycobacterial adaptation to rapidly cooling conditions, and thus contribute to their well-known versatility and resilience in both natural and artificial environments.

## Methods

**Detection of the MMP gene cluster in actinobacterial genomes.** The protein sequences of the proposed MMP biosynthetic gene cluster[24] (MeT1: WP 110570796.1 ortholog MHAS_04405, MSMEG_6482 and MAV_2758; ManT: WP 005631138.1 ortholog MHAS_04402, MSMEG_6484 and MAV_2755; pMeT3: WP 005631136.1 ortholog MHAS_04403, MSMEG_6483 and MAV_2756 and MmpH: WP 018354040.1 ortholog MHAS_04404, MSMEG_6481 and MAV_2757) were retrieved from the *M. hassiacum* genome[40] at the NCBI database and independently used in BLAST searches (https://blast.ncbi.nlm.nih.gov/Blast.cgi) against mycobacterial genomes and those of eight closely related species of actinobacteria in which either MMP or MGLP had been detected[41,42], as well as against the genus *Hoyosella* of the Mycobacteriaceae family[43]. Sequences with more than 25% amino acid sequence identity (considered of low to moderate similarity) were included in the alignments and those with more than 50% identity and 70% coverage (considered significantly similar) were used to search for the protein cluster in actinobacterial genomes. The phylogenetic relationship for the complete genomes of 162 mycobacterial species and related actinobacteria (Table S1) was inferred using the bcgTree pipeline[44]. Homology was detected using HMMER v3.3[45], from each actinobacterial proteome against hidden Markov model (hmm) profiles of 107 essential single copy genes found in more than 95% of all known bacteria[44]. For each gene, best matches were retained and multiple query sequences aligned using

MUSCLE v3.8.31[46]. Alignments were refined by selection of conserved blocks with Gblocks 0.91b[47] and a maximum likelihood phylogenetic tree was built with RAxML v7.7.2[48]. When required, genomes were compared to determine the Average Nucleotide Identity (ANI) and Genome to Genome Distance (GGD) pairwise values using the OrthoANIu algorithm[49] and the Formula 2 option of the Genome to Genome Distance Calculator 2.1[50]. Genome pairs with ANI > 95% or GGD > 70% were considered of the same species. Genomes with ANI ≤ 97% and GGD ≤ 80% were considered subspecies[51].

**Bacterial strains and culture conditions.** *Mycobacterium hassiacum* (DSM 44199) and *Mycobacterium avium* (DSM 44156) were obtained from the German Collection of Microorganisms and Cell Cultures (Germany) and *Mycobacterium smegmatis* mc²155 (ATCC 700084) from LGC Standards (Spain). *Mycobacterium phocaicum* was from our collection (isolated from hospital facilities)[52] (Table S2). *Mycobacterium hassiacum* and *M. smegmatis* were grown in a glycerol-based medium (GBM) with 20 g/L glycerol, 5 g/L casamino acids (Difco), 1 g/L fumaric acid, 1 g/L K₂HPO₄, 0.3 g/L MgSO₄, 0.02 g/L FeSO₄ and 2 g/L Tween 80, at pH 7.0[53]. *Mycobacterium avium* was grown in Middlebrook 7H10 plates. *Escherichia coli* DH5α and BL21 (DE3) strains were cultured in LB medium supplemented with the required antibiotics. Electrocompetent *M. smegmatis* mc²155 cells were prepared as previously described[54].

**Extraction and purification of mycobacterial MMP.** MMP was extracted from four mycobacterial species: *M. smegmatis* for its ease of manipulation; *M. hassiacum* for being the source of the enzymes studied in this work; *Mycobacterium phocaicum* for being an example of an RGM lacking the MMP gene cluster; and *M. avium* for being an important pathogenic SGM containing the MMP cluster. MMPs were extracted and purified using a modification of a published protocol[10,55]. Cells were grown in GBM to late exponential growth phase (OD₆₀₀ = 3.0). Total lipids and amphiphilic PMPS were extracted from 30 g cell biomass with chloroform:methanol (1:2) for 3 h under constant agitation. The mixture was centrifuged (8000 × g, 30 min) and the supernatant evaporated in a rotary evaporator, prior to a second rapid extraction step with chloroform:methanol (2:1). The aqueous phase was concentrated by evaporation, loaded onto a Resource RPC column (GE Healthcare) equilibrated with water, and eluted with 30% (v/v) methanol at a flow rate of 3 mL/min. The eluted fractions were analyzed on a glass-backed silica-gel 60 coated TLC plate without indicator (Merck) using chloroform:methanol:water (55:40:10, v/v/v) as mobile phase. MMP-containing fractions were detected by spraying with α-naphthol-sulfuric acid solution followed by charring at 120 °C[56]. Residual methanol was evaporated in a centrifugal vacuum concentrator (Thermo Fisher) and fractions containing MMP were pooled and further purified by size-exclusion chromatography on a HiPrep 16/60 Sephacryl S-200 HR column (GE Healthcare) equilibrated with water. Eluted fractions were analyzed by TLC as above and those containing MMP were pooled and lyophilized. For rapid detection of MMP in different mycobacteria, a miniaturized protocol starting from 4 mL of bacterial culture was developed. Following the two consecutive extractions with chloroform:methanol, the aqueous phase was concentrated by evaporation, loaded onto 3 mL HyperSep C18 SPE cartridges (Fisher Scientific), eluted with 40, 50 or 80% (v/v) methanol and the eluted fractions were analyzed by TLC. MMPs extracted with both methods were dissolved in 50% (v/v) acetonitrile, 0.1% (v/v) formic acid and analyzed by MS in a Triple TOF™ 5600 or Triple TOF™ 6600 System (Sciex) by direct infusion (10 μL/min flow rate) and the ionization source (ESI DuoSpray™ Source) operated in either the positive (ion spray voltage of 5500 V) or negative mode (ion spray voltage of −4500 V) and set at 25 psi both for nebulizer gas 1 (GS1) and curtain gas (CUR). The acquisition was performed in full scan mode (TOF-MS mode) and product ions were obtained using collision energy ramping. All masses indicated in spectra are monoisotopic masses.

**Protein production and purification.** MmpH. A synthetic gene coding for *M. hassiacum* MmpH and sequence-optimized for expression in *E. coli* (GenScript) was cloned into the NdeI and HindIII restriction sites of the pET30a vector (Novagen) and C-terminal His6-tagged MmpH was overexpressed in *E. coli* BL21(DE3) cells. This vector was used as template to obtain MmpH variants Asp47Ala, Asp50Ala and Glu262Ala by the QuikChange site-directed mutagenesis method with primers V1 to V6 (Table S3). Cells were grown in LB medium with 50 μg/mL kanamycin at 37 °C until OD₆₀₀ ≈ 0.8, when the incubation temperature was decreased to 25 °C and recombinant protein expression induced by addition of 0.5 mM isopropyl β-D-1-thiogalactopyranoside and allowed to proceed for 12–14 h. The cells were harvested by centrifugation (3000 × g, 20 min, 4 °C), resuspended in 20 mL buffer A (20 mM sodium phosphate pH 7.4, 0.5 M NaCl, 20 mM imidazole) and stored at −20 °C. The cell suspensions were thawed (60 min on ice) in the presence of lysozyme and DNAse I (each at 5 μg/mL final concentration), 5 mM MgCl₂, 1 mM PMSF, and EDTA-free protease inhibitor cocktail (Roche). Cells were disrupted by sonication on ice with three 20 s-long 40 Hz pulses (10 s pause between pulses) per 7 mL of lysate. After centrifugation (17000 × g, 40 min, 4 °C), the clarified supernatant was filtered through a low protein-binding 0.45 μm pore syringe filter (Millipore) and loaded onto a 5 mL HisTrap HP column (GE Healthcare), previously equilibrated with buffer A. Bound

proteins were eluted with 500 mM imidazole in buffer A and their purity assessed by SDS-PAGE analysis. MmpH-containing fractions were pooled and concentrated on a centrifugal ultrafiltration device (30 kDa molecular weight cut-off (MWCO); Millipore). MmpH used for crystallization was further purified on a HiPrep Sephacryl S-200 HR 16/60 size exclusion chromatography column (GE Health-care), equilibrated with 20 mM bis-tris propane (BTP) pH 7.5, 200 mM NaCl. MmpH-containing fractions were concentrated and concomitantly equilibrated in 20 mM BTP pH 7.5, 100 mM NaCl on a centrifugal ultrafiltration device (30 kDa MWCO; Millipore). All MmpH variants were produced and purified using the protocol followed for the wild-type enzyme. Selenomethionyl MmpH was over-expressed in *E. coli* B834 (DE3). The cells from an overnight pre-culture (37 °C) in LB medium were pelleted, washed thrice in the same volume of sterile water, resuspended in 1 mL of water, and used to inoculate a commercial medium (Molecular Dimensions) containing 40 µg/mL L-selenomethionine, 50 µg/mL kanamycin. The culture was then incubated at 37 °C until $OD_{600} \approx 0.8$. Purification of selenomethionyl MmpH followed the protocol developed for the native protein, except for the inclusion of 1 mM DTT in all steps. Pure protein was flash-frozen in liquid nitrogen and stored at −80 °C until needed. Protein concentrations were estimated from the absorbance of the samples at 280 nm.

ManT. A synthetic gene coding for *M. hassiacum* ManT and sequence-optimized for expression in *E. coli* (GenScript) was cloned into the pETM11 vector (Novagen) using NcoI and EcoRI restriction sites. For ManT variants Glu313Ala, Glu321Ala and Lys240Ala, commercial synthetic genes sequence-optimized for expression in *E. coli* (NZYTech) were cloned into the pETM11 vector, as for the wild-type gene. The resulting N-terminal His6-tagged ManT was overexpressed in *E. coli* BL21(DE3) cells. Overexpression, cell lysis and affinity chromatography were performed as described for MmpH except for using buffer B (20 mM sodium phosphate pH 8.0, 0.5 M NaCl, 5 mM β-mercaptoethanol, 20 mM imidazole, 0.1% (v/v) Triton X-100). Fractions eluted with 500 mM imidazole in buffer B and containing recombinant ManT, as assessed by SDS-PAGE analysis, were pooled, concentrated on a centrifugal ultrafiltration device (30 kDa MWCO; Millipore) and loaded onto a HiPrep Sephacryl S-300 HR 26/60 size exclusion chromatography column (GE Healthcare), equilibrated with 20 mM sodium phosphate pH 8.0, 0.5 M NaCl, 1 mM DTT, 0.1% (v/v) Triton X-100. ManT-containing fractions were concentrated, flash-frozen in liquid nitrogen and stored at −80 °C until used. Protein concentration was estimated by the Bradford assay or from the absorbance at 280 nm. All ManT variants were produced and purified using the protocol followed for the wild-type enzyme.

MeT1. A synthetic gene coding for *M. hassiacum* MeT1 was sequence-optimized (GenScript) for expression in *E. coli*, cloned into the NdeI and HindIII restriction sites of the pET30a vector (Novagen), and overexpressed in *E. coli* BL21(DE3) cells as a C-terminal His6-tagged MeT1. Overexpression, cell lysis and affinity chromatography were performed as described for MmpH. After affinity chromatography MeT1 was pooled and dialyzed against 10 mM BTP pH 7.5, 50 mM NaCl, concentrated using a 10 kDa 10 kda MWCO centrifugal ultrafiltration device (Millipore), flash-frozen in liquid nitrogen, and stored at −80 °C[24].

**Synthetic oligomannosides**. The reactions and protocols required for the synthesis of propylated 4α-mannotriose (sMan₃), 3,3',3''-tri-*O*-methyl-4α-mannotriose (sMetMan₃), 4α-mannotetraose (sMan₄) and 3,3',3'',3'''-tetra-*O*-methyl-4α-mannotetraose (sMetMan₄) are described in detail in the Supplementary Methods and their schematic synthesis are also presented. For each synthetic compound, analytical TLC was performed in aluminum-backed silica gel Merck 60 F254. After most reactions, the products were purified by preparative flash column chromatography using silica gel Merck 60H. Reagents and solvents were purified and dried according to the literature[57]. All $^1$H-NMR spectra were obtained at 400 MHz in CDCl₃, MeOH-d4 or D₂O with chemical shift values (δ) in ppm downfield from tetramethylsilane in the case of CDCl₃, and $^{13}$C-NMR spectra were obtained at 100.61 MHz in CDCl₃, MeOH-d4 or D₂O. Assignments are supported by 2D correlation NMR studies. The specific rotations ($[\alpha]_D^{20}$) were measured using an automatic polarimeter. NMR spectra for all novel compounds reported are provided in Supplementary Methods.

**Enzyme substrate specificities**. MmpH. The substrate specificity of MmpH was assessed using maltotetraose, maltopentaose, maltohexaose, maltoheptaose and maltooctaose (all purchased from Sigma), synthetic oligo mannosides sMan₃, sMetMan₃, sMan₄ and sMetMan₄ (all synthesized in this work), 4β-mannotriose, 4β-mannotetraose and β-(1→4)-mannan (all purchased from Megazyme), purified mycobacterial MMP (see above) and MGLP (byproduct of the purification of MMP). Reaction mixtures containing MmpH (5.1 µM), 25 mM BTP pH 7.5 and 2.5 mM sugar were incubated at 37 °C for 2 h. Product formation was monitored by TLC with a chloroform:methanol:water (55:40:10, v/v/v) solvent system and developed with α-naphthol-sulfuric acid [56]. Product formation with MMP was assessed after 5-, 15-, and 30 min incubation. All products formed were purified and analyzed by MS (see above).

ManT. The substrate specificity of ManT was assessed using guanosine 5'-diphospho-D-mannose (GDP-Man) (Sigma). D-glucose, D-mannose (Man), 3-*O*-methyl-D-glucose, methyl D-glucose, methyl D-mannose, maltose and maltotetraose (all from Sigma), maltotriose (Carbosynth), 4α-mannobiose (Man₂,

Dextra), 4β-mannobiose (Carbosynth), 4β-mannotriose and 4β-mannotetraose (both from Megazyme), the chemically-synthesized 3-*O*-methyl-mannose (sMetMan), 3,3'-di-*O*-methyl-4α-mannobiose (sMetMan₂), 1,3,3'-tri-*O*-methyl-4α-mannobiose (sMet₁,₃Man₂) obtained previously[24], the sMan₃, sMetMan₃, sMan₄, sMetMan₄ synthesized in this work, and the four different-order natural mannoligosaccharides obtained from MMP hydrolysis by MmpH (see below) were tested as possible ManT substrates. Reaction mixtures containing pure ManT (2.1 µM), 25 mM BTP pH 7.5, 5 mM donor substrate and 5 mM acceptor substrate were incubated at 37 °C for 2 h. Product formation was monitored by TLC on silica gel 60 plates (Merck) with a butanol:ethanol:water (4:4:2, v/v/v) solvent system and stained with α-naphthol-sulfuric acid solution, followed by charring at 120 °C[56]. Products formed were purified and analyzed by MS (see above).

**Enzymatic assays**. MmpH quantification assay. MmpH activity data were obtained using a sensitive reducing-sugar *p*-hydroxybenzoic acid hydrazide (pHBAH)-based discontinuous assay[58,59]. A pHBAH working solution was freshly prepared by diluting a pHBAH stock solution (5% (w/v) in 0.5 M HCl) 1:4 with 0.5 M NaOH (v/v)[58,59]. Assays were initiated by addition of 1.3 µM MmpH to buffered reaction mixtures containing 1.5 mM MMP and stopped by rapid cooling on an ethanol-ice bath. The enzyme was inactivated by the addition of pHBAH working solution (150 µL), followed by 100 mM BTP pH 7.5 (25 µL). The mixtures were incubated 5 min at 95 °C and their absorbance at 415 nm registered in 96-well microtiter plates. The effect of pH was determined at 45 °C in 50 mM MES (pH 5.5 to 6.5) or BTP (pH 6.5 to 8.5) buffer and the temperature profile were determined between 25 and 65 °C in 50 mM BTP pH 7.5. The influence of divalent cations on enzyme activity was examined by incubating the reaction mixtures at 37 °C for 30 min with 2.5 mM of the chloride salts of $Mg^{2+}$, $Mn^{2+}$, $Ca^{2+}$, $Cu^{2+}$, $Fe^{2+}$, $Co^{2+}$ and $Zn^{2+}$ and without added cations, or in the presence of 10 mM EDTA. These reactions were analyzed by TLC (see above). All experiments were performed in triplicate.

MmpH kinetic parameters. Kinetic parameters for MmpH were determined at 37 and 45 °C using the discontinuous assay described above. The $K_m$ and $V_{max}$ values for MMP were determined in reaction mixtures containing 50 mM BTP pH 6.5 and 1.3 µM enzyme. All experiments were performed in triplicate. Kinetic parameters were calculated with Prism v. 5.00 (GraphPad).

ManT quantification assay. The enzymatic activity of ManT was probed using a discontinuous method through indirect quantification of the GDP released from GDP-Man[60]. Assays to determine pH and temperature profiles and the effect of divalent cations, of sodium chloride and detergent (Triton X-100) concentration on enzyme activity were initiated by the addition of ManT (0.52 µM) to reaction mixtures (25 µL) containing the appropriate buffer (50 mM), 1.5 mM sMetMan₄, 2.5 mM GDP-Man, 5 mM MgCl₂, 0.1% (v/v) Triton X-100 and 25 mM NaCl. The reaction was stopped by cooling on ethanol-ice and the enzyme inactivated by the sequential addition of 5 N HCl and 5 N NaOH (1.25 µL each). Freshly prepared quantification mixture (72.5 µL of 2 mM phosphoenolpyruvate (PEP), 2.5 mM KCl, 800 µM NADH (Sigma) and 1 unit (U) each of pyruvate kinase and lactate dehydrogenase (PK/LDH, Sigma)) was added, the reactions were incubated at 37 °C for 15 min and the absorbance at 340 nm registered. The effect of pH was determined at 37 °C in 50 mM BTP (pH 6.5 to 9.0) or CAPSO (pH 9.0 to 10.0) buffers and the temperature profile was determined between 20 and 55 °C in 50 mM BTP pH 8.5. The effect of divalent cations on enzyme activity was examined by incubating the reaction mixture for 30 min at 37 °C with 5 mM of the chloride salts of $Mg^{2+}$, $Mn^{2+}$, $Ca^{2+}$, $Cu^{2+}$, $Fe^{2+}$, $Co^{2+}$, $Zn^{2+}$ and without added cations, or in the presence of 10 mM EDTA, and analyzed by TLC as described above. The effect of MgCl₂ (0.1 to 20 mM), NaCl (0 to 50 mM) and Triton X-100 (0 to 0.5% (v/v)) on ManT activity was also evaluated. All experiments were performed in triplicate.

ManT kinetic parameters. The kinetic parameters for ManT activity were determined at 37 °C using a continuous method. Activity assays were performed in 50 mM BTP pH 8.5, 7.5 mM MgCl₂, 5 mM NaCl, 0.1% (v/v) Triton X-100, 2 mM PEP, 800 µM NADH, 2.5 mM KCl and 1 U PK/LDH (Buffer C) in 96-well microtiter plates. Reactions were initiated by the addition of enzyme (0.52 µM) and product formation was indirectly quantified by monitoring at 340 nm the amount of GDP released. The $K_m$ and $V_{max}$ values for GDP-Man, sMetMan₄ or sMan₄ were determined using fixed saturating concentrations of donor (2.5 mM and 5 mM GDP-Man) or of acceptor (1.5 mM sMetMan₄ or 2.5 mM sMan₄) and a varying concentration of acceptor or donor, respectively. For the E313 variant, to determine $K_m$ and $V_{max}$ values for GDP-man and sMetMan4, 3.5 mM of sMetMan4 and 2.5 mM of GDP-Man were used as fixed substrate, respectively. All experiments were performed in triplicate with appropriate controls to exclude the effect of substrate degradation. Kinetic parameters were calculated with Prism v. 5.00 (GraphPad Software).

**Testing the oligomannoside products of MmpH activity as ManT and MeT1 substrates**. The activity of ManT using either the synthetic or the natural oligomannosides (**a-d**) obtained from MMP hydrolysis by MmpH (see above) was quantified with the continuous assay (see above; reaction mixtures containing 1.5 mM acceptor oligomannoside and 2.5 mM GDP-Man in Buffer C). The activity of MeT1 using the synthetic or the natural oligomannosides at 37 °C for 2 h in 25 mM BTP pH 7.5, 5 mM MgCl₂, 1.5 mM donor *S*-adenosyl-methionine (SAM),

2.5 mM oligomannoside and 3.2 μM MeT1 was qualitatively analyzed by TLC and quantified from the *S*-adenosyl-homocysteine (SAH) released from SAM through a continuous, previously described[24] method. Reactions were initiated by the addition of enzyme (1.9 μM) to mixtures containing 50 mM BTP pH 7.5, 1.5 mM acceptor substrate, 1 mM SAM, 100 μM MnCl$_2$, 500 μM DTNB, 500 μM NAD$^+$ and 1.7 μM SahH. The release of SAH was monitored at 412 nm[24]; all experiments performed at least in triplicate.

**Mass spectrometry analyses.** Reactions containing MmpH (50 mM BTP pH 6.5, 1.5 mM purified MMP; 2 mL volume) were incubated for 4 h at 45 °C, loaded onto a Resource RPC column (GE Healthcare) and eluted at 3 mL/min with 30% (v/v) methanol in water. The purest fractions, as assessed by TLC (see above), were pooled and vacuum dried for further MS analyses. ManT products, obtained as described above, were purified by TLC[61]. Briefly, the reaction mixtures were spotted onto TLC plates and separated with a butanol:ethanol:water (4:4:2, v/v/v) solvent system. The relevant spots were identified by staining the marginal lanes of the TLC plates. The corresponding regions in the unstained lanes of the plate were scraped, each product was extracted from the silica with ultrapure water and analyzed by LC-MS (see above).

**Crystallization, data collection and processing.** Initial crystallization conditions for native MmpH and ManT were screened at 20 °C using the vapor diffusion sitting-drop method with commercial factorial crystallization screens. Drops consisting of equal volumes (1 μL) of protein at 9 mg/mL (MmpH) or 12 mg/mL (ManT) and precipitant solution were equilibrated against a 300 μL reservoir.

MmpH. Monoclinic MmpH crystals belonging to space group P2$_1$ were obtained after 10 days using 0.1 M Tris-HCl pH 8.5, 0.02 M MgCl$_2$, 20–22% (w/v) polyacrylic acid as precipitant. Before data collection, the crystals were cryoprotected by brief immersion in 0.1 M Tris-HCl pH 8.5, 0.02 M MgCl$_2$, 26% (w/v) polyacrylic acid, 20% (v/v) glycerol and flash-cooled in liquid nitrogen. Orthorhombic (space group P2$_1$2$_1$2$_1$) selenomethionyl MmpH crystals were obtained after 2 days from drops consisting of equal volumes (1 μL) of protein (at 3 mg/mL) and precipitant solution equilibrated against 300 μL of 0.1 M Na-HEPES pH 7.5, 15–17% (w/v) PEG 8000. Prior to data collection, the selenomethionyl MmpH crystals were cryoprotected by brief immersion in 0.1 M Na-HEPES pH 7.5, 21% (w/v) PEG 8000, 10% (v/v) glycerol and flash-cooled in liquid nitrogen. All X-ray diffraction data were collected from cooled (100 K) crystals (one crystal per data set). The native MmpH data (1440 images in 0.25° oscillation steps and 0.25 s exposure) were recorded on a PILATUS 6 M (Dectris) detector using a wavelength of 0.9809 Å at beamline BL13-XALOC[62] of the ALBA Synchrotron (Cerdanyola del Vallès, Spain). The selenomethionyl MmpH data (360 images in 1° oscillation steps and 8 s exposure) were recorded on a Quantum 315r CCD detector (ADSC) using a wavelength of 0.9799 Å at beamline BM30A[63] of the European Synchrotron Radiation Facility (Grenoble, France). All diffraction data were integrated with XDS[64] and reduced with utilities from the CCP4 program suite[65]. Data collection statistics are summarized in Table S4.

ManT. Monoclinic ManT crystals belonging to space group C2 were obtained after 3 days using 0.1 M Na-HEPES pH 7.5, 0.2 M sodium acetate, 25–29% (w/v) PEG 3350 as precipitant. Prior to data collection, the crystals were cryoprotected by brief immersion in 0.1 M Na-HEPES pH 7.5, 0.2 M sodium acetate, 25% (w/v) PEG 3350, 10% (v/v) glycerol and flash-cooled in liquid nitrogen. X-ray diffraction data for ManT were collected from a single cooled (100 K) crystal (800 images in 0.25° oscillation steps and 0.38 s exposure), recorded on a PILATUS 6 M (Dectris) detector using a wavelength of 1.0332 Å at beamline BL13-XALOC[62] of the ALBA Synchrotron (Cerdanyola del Vallès, Spain), and processed with iMOSFLM[66] and AIMLESS[67]. Data collection statistics are summarized in Table S4. The diffraction images were deposited with the SBGrid Data Bank[68]: https://doi.org/10.15785/SBGRID/874 and https://doi.org/10.15785/SBGRID/875 (MmpH native and Se-Met data sets, respectively) and https://doi.org/10.15785/SBGRID/873 (ManT).

**Structure determination.** MmpH. The structure of MmpH was solved by single-wavelength anomalous diffraction using the anomalous signal of selenium with the SHELXC/SHELXD/SHELXE pipeline[69] and the HKL2MAP GUI[70]. Automated model building with ARP/wARP[71] on the experimental 2.01 Å resolution map docked in sequence 666 of 746 residues. The ARP/wARP model was used for molecular replacement (MR) on the high-resolution native dataset using PHASER[72].

ManT. The structure of ManT was solved by MR using the coordinates of PimA (PDB entry 4N9W[73]). The region encompassing residues 239-410 of the mixed atoms model generated by SCWRL[74] was used as search model with PHASER[72]. After the first round of refinement with PHENIX[75], R factors were still above 0.5. The output model from PHASER was then subjected to morphing and automated model building with Autobuild[76], leading to a substantial improvement of the electron density maps, accompanied by improved statistics ($R_{work} = 0.33$ and $R_{free} = 0.37$) after partial completion of the model.

The molecular models of both MmpH and ManT were iteratively improved in alternating cycles of manual model building with COOT[77] and of refinement with PHENIX[75] until completion (refinement statistics are summarized in Table S4). The final MmpH model comprises residues 6–351 (molecule A) and 4–348

(molecule B). The final ManT model comprises residues 1–67, 77–90, 93–152, 166–233, 239–266, and 270–408 (molecule A) and 1–16, 18–39, 45–49, 93–105, 118–121, 134–138, 140–150, 177–235 and 238–405 (molecule B). Although the engineered N-terminal affinity tag was not removed during purification, only three residues (Gln-Gly-Ala) could be modeled in molecule A and one (Ala) in molecule B. Refined coordinates and structure factors for MmpH and ManT were deposited at the PDB with accession numbers 7QSJ and 7QSG, respectively. All crystallographic software was supported by the SBGrid project[78].

**Construction of an MmpH-negative mutant.** An *M. smegmatis* mc$^2$155 Δ*mmpH* mutant strain (*MsmegΔmmpH*) was generated using the allelic exchange method based on the pGOAL/pNIL system[79]. The p2NIL and pGOAL19 plasmids were obtained from Addgene (plasmids # 20188 and # 20190, respectively). PCR reactions were performed using KOD hot start DNA polymerase (Merck Millipore) and ligation was accomplished using ExpressLink T4 DNA Ligase (Invitrogen). *M. smegmatis* mc$^2$ 155 was grown in GBM medium[53] and electrocompetent cells were prepared as described[54] using GBM medium. The p2NIL-Δ*mmpH* plasmid was generated stepwise by amplification of an 877 bp fragment upstream of the target deletion, using oligonucleotide primers A1 and A2 and WT *M. smegmatis* mc$^2$ 155 genomic DNA as template (Table S3). The resulting PCR product was inserted into p2NIL using the restriction enzymes KpnI and BamHI to generate p2NIL-F1. A second fragment, located 1357 bp downstream of the target deletion in the *mmpH* gene, was amplified using oligonucleotide primers A3 and A4 and inserted into p2NIL-F1 using BamHI and HindIII to generate p2NIL-Δ*mmpH*. Hence, the construct p2NIL-Δ*mmpH* carries a 678 bp sequence upstream of the *mmpH* gene plus the initial 195 bp and the last 589 bp of *mmpH* (corresponding to a 290 bp deletion), followed by a 761 bp segment downstream of *mmpH* (includes p*metT3*). The pGOAL19 selectable marker cassette carrying the *sacB*, *lacZ* and *hyg* genes was introduced into linearized p2NIL-Δ*mmpH* using the PacI restriction site[79] originating p2NIL-Δ*mmpH*-sel. The resulting construct was confirmed by restriction analysis and DNA sequencing using primers A1 and A4 (Table S3). The p2NIL-Δ*mmpH*-sel suicide vector was treated with UV light (100 mJ/cm$^2$, Stratagene Stratalinker 1800)[79] and introduced into *M. smegmatis* mc$^2$ 155 electrocompetent cells by electroporation (Biorad MicroPulser Electroporation System; EC2 program). Transformants were selected on GBM agar plates supplemented with hygromycin (200 mg/mL), kanamycin (25 mg/mL) and X-Gal (50 mg/mL). Blue colonies were re-streaked onto GBM agar plates and incubated at 35 °C. A loopful of cells was then resuspended in GBM broth, serially diluted, and plated on solid medium with 2% (w/v) sucrose and X-Gal (50 μg/mL). White sucrose-resistant colonies were selected for patch testing for kanamycin sensitivity and disruption of the *mmpH* reading frame. For white KanS, HygS and SucR colonies the disruption of the reading frame was confirmed by PCR and DNA sequencing using primers A1 and A2, A8 and A2, A3 and A7, A6 and A5 (Table S3).

**Characterization of the MMP-negative *M. smegmatis* mutant.** The production of polymethylated polysaccharides (PMPS) was examined in the *MsmegΔmmpH* mutant using WT *M. smegmatis* mc$^2$155 as control strain. PMPS were extracted from cells grown in GBM at 37 °C and examined by TLC as described above. Cells in exponential and stationary phases were evaluated for overall morphology by standard microscopy (100-fold magnification). To test the response of *MsmegΔmmpH* to thermal stress, mutant and WT were grown at temperatures between 15 and 45 °C in GBM media. Cultures (125 mL) were incubated with continuous shaking (160 rpm) in 300 mL Erlenmeyer flasks. Cultures were inoculated to an initial OD$_{600}$ of 0.05 and growth was measured by monitoring the increase of OD$_{600}$.

**Statistics and reproducibility.** Statistical analyses of enzyme kinetic data (Figs. 2G, H, 4K–N, and 5I, J) were performed using Prism v. 5.01 (GraphPad software), while. R v. 4.0.4 was used for other statistical analyses. Quantitative data were expressed as mean ± SD or mean ± SEM (Fig. 6D). Reproducibility was confirmed by performing at least three independent determinations. Enzymatic data reproducibility was confirmed using different enzyme batches (Fig. 2E–H and Fig. 4F–N).

**Reporting summary.** Further information on research design is available in the Nature Portfolio Reporting Summary linked to this article.

## Data availability

All data are available in the main text or in the Supplementary Information file. Source data for figures can be found in Supplementary Data 1. Refined coordinates and structure factors for MmpH and ManT were deposited at the Protein Data Bank with accession numbers 7QSJ and 7QSG, respectively.

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

## Acknowledgements

This work was funded by Portuguese national funds via FCT—Fundação para a Ciência e a Tecnologia through projects PTDC/BIA-MIC/0122/2021, UIDB/04539/2020, UIDP/04539/2020 and LA/P/0058/2020; through PhD Fellowship SFRH/BD/101191/2014 (to M.C.); through contract POCI-01-0145-FEDER-029221 (to A. M.); through contract DL 57/2016/CP1355/CT0011 (to J.R.-R.) and by the European Social Fund through Programa Operacional Capital Humano in the form of Postdoctoral Fellowship SFRH/BPD/108004/2015 (to J.R.-R.) and by the European Regional Development Fund (FEDER) through the COMPETE 2020-Operational Programme for Competitiveness and Internationalization (POCI), Portugal 2020 in the form of grant POCI-01-0145-FEDER-029221, and the National Mass Spectrometry Network (RNEM) under the contract POCI-01-0145-FEDER-402-022125 (ref.: ROTEIRO/0028/2013). M.R.V. acknowledges MostMicro Research Unit, financially supported by LISBOA-01-0145-FEDER-007660 funded by FEDER funds through COMPETE2020 (POCI) and by national funds through FCT. The NMR data was acquired at CERMAX, ITQB-NOVA, Oeiras, Portugal with equipment funded by FCT, project AAC 01/SAICT/2016. We thank Igor Tiago (Centre for Functional Ecology, University of Coimbra) for support with computation and bioinformatic analyses. We thank ALBA Synchrotron (Cerdanyola del Vallès, Spain) and the European Synchrotron Radiation Facility (Grenoble, France) for provision of synchrotron radiation facilities, and their staff for help with data collection. The support of the X-ray Crystallography scientific platform of i3S (Porto, Portugal) is also acknowledged.

## Author contributions

N.E., P.J.B.P., A.M., J.R.-R., and M.R.V. conceptualized the study or designed the experiments. A.M., M.C., J.R.-R., J.A.M., V.M., and V.M.M performed the experiments. A.M., M.C., J.R.-R., J.A.M., S.M.-R., M.R.V., P.J.B.P., and N.E. performed data analyses. N.E., P.J.B.P., M.R.V., and B.M. contributed protocols, reagents and materials. A.M., M.C., J.R.-R., M.R.V., P.J.B.P., and N.E. wrote the manuscript. All authors reviewed and approved the final version of the manuscript.

## Competing interests

The authors declare no competing interests.
