## [Peer Review File · Communications Biology]

Reviewers' comments:

Reviewer #1 (Remarks to the Author):

In this manuscript, Maranha et al. reported the novel recycling biosynthetic mechanism of 3-O-methylmannose polysaccharide (MMP) in nontuberculous mycobacteria (NTM). The authors' group previously revealed the structure and enzymatic features of MeT1 which is one of the key enzymes in the MMP biosynthetic pathway. They here characterized the structures and enzymatic properties of other key two enzymes, MMP hydrolase (MmpH) and mannosyltransferase (ManT). Most strikingly, the authors found that the products of MmpH were used as substrates for ManT and mannose 1-O-methyltransferase (MeT1), serving as a key for this recycling pathway. Overall, the study was well conducted, and the paper is well written. After several concerns will be addressed, I am happy to support publication of this study in *Commun Biol*. Specific comments are as follows.

Major comments

1. Regarding the structure of MmpH, even without ligand-complex crystal, the authors nicely discussed and predicted the substrate recognition mechanism by comparing with structurally related G1d and CpGH125. Such prediction should be experimentally confirmed using the mutant enzymes which lack any key putative residue.
2. The authors detected sequential addition of up to 3 Man residues by ManT reaction in vitro. What would be the number of Man residues with longer incubation time or excess amount of the enzyme? This is an important point to estimate the biosynthetic mechanism of MMP.
3. Line 192: The authors described that MmpH showed the higher apparent K_m value at 45 degree than 37 degree. The higher V_{max} is well consistent with Fig. 2E, but I feel that the K_m values are only slightly different and very similar between 45 and 37. What do the authors want to claim from the "higher apparent K_m "?
4. Although MmpH and ManT were the enzymes in the same pathway, Fig. 2 and 4 showed that the two recombinant enzymes derived from the same species have different optimal pH and degree. This should be discussed more carefully.
5. Introduction. Previous findings about the MMP biosynthetic pathway are nicely summarized, but information about the putative 3-O-methyltransferase (pMeT3) is lacking. I would suggest that the putative roles or related information of pMeT3 are also described.

Minor points

6. Line 277: A blank line better be inserted before the title.
7. Line 36: "endomannosidade" should be "endomannosidase"
8. Line 174: "purify" should be corrected.
9. Line 181-184: "c" and "d" should be bold.
10. Line 297-298: "a" and "d" should be bold.

Reviewer #2 (Remarks to the Author):

The manuscript entitled "Self-recycling and partially conservative replication of mycobacterial methylmannose polysaccharides" by Maranha, Costa, Ripoll-Rozada and colleagues studied the mycobacterial pathway involved in recycling and replication of 3-O-methylmannose polysaccharide (MMP). Using bioinformatics and phylogenetics, the authors describe the distribution of MMP biosynthetic gene clusters in mycobacterial genomes and found a wide distribution of these clusters in nontuberculous mycobacteria (NTM), including the first description of such genes in slowly growing

organisms. They also biochemically and structurally characterized two enzymes important for MMP biosynthesis, an endo- α -(1 \rightarrow 4)-mannosidase that hydrolyses MMP and an α -(1 \rightarrow 4)-mannosyltransferase, the first protein described to be able to elongate a polysaccharide polymer with α -(1 \rightarrow 4)-linked mannoses. The endomannosidase products can be used as substrates for elongation of new MMP chains by the mannosyltransferase and by a methyltransferase coded by the same gene cluster. This study also provides evidence that MMP could be linked to mycobacterial cold adaptation. The authors conclude their study proposing an elegant mechanism of MMP biogenesis relying on pre-existing mature MMP and on the concerted action of the endomannosidase, the mannosyltransferase and two additional specific methyltransferases. In summary, the manuscript is a valuable contribution to the literature of biogenesis and recycling of mycobacterial polysaccharides.

The manuscript is well written, the work is technically sound and may be published after minor revisions addressing the points below:

- 1) All phylogenetic trees do not present bootstrap values. Please provide bootstrap or other branch support value to the phylogenetic trees.
- 2) The authors searched mycobacterial genomes looking for a 4-gene cluster involved in MMP biosynthesis, indicating extensive and scattered distribution of these genes across mycobacteria. But the fact that this cluster is also found in *Gordonia rubripertincta* and three *Nocardia* (*N. farcinica*, *N. otitidiscaviarum* and *N. pseudobrasiliensis*; Fig. 1B) suggests the distribution is even more widespread than the authors currently describe in the manuscript and the same pattern of extensive and scattered distribution could happen at least in the order *Corynebacteriales*. It would be interesting to analyze the broader distribution of MMP clusters in *Corynebacteriales* / *Actinomycetia*.
- 3) There are two references to "data not shown" in the manuscript (lines 160 and 310). All relevant data should be shown and provided in the manuscript (e.g. as supplementary material) or references to these data should be entirely removed from the study.
- 4) The manuscript nicely describes specific mutations affecting the activity of ManT, but not of MmpH. There are a few positions that seem to be essential for the MmpH function, such as D47, D50, E262, and residues lining a potential subsite occupied by glycerol. The confirmation of which residues are important for the catalytic activity and sugar binding of MmpH would strengthen the work.
- 5) It is interesting to notice that the ManT E313A mutant, although having lower enzymatic activity (Fig. 5E), seems to be able to produce sP1 (pentamannoside) and sP2 (hexamannoside) products, but not sP3 (heptamannoside), in contrast to the Wt protein (Fig. 5F). What would be an explanation for this inability of the E313A mutant to produce heptamannosides?
- 6) Conservation analysis and electrostatic surface potential could be mapped onto the ManT structure, similarly to the analyses shown in Figures 3C-D for MmpH.
- 7) The source of the enzymes studied in this work is *M. hassiacum*, a thermophilic organism that grows optimally at 50 °C and up to 65 °C. However, the optimal activities of ManT and MmpH are in the 40 - 45 °C range. The manuscript provides clues linking MMP to cold adaptation; is it possible that there is a relationship between the smaller thermostability of these enzymes and *M. hassiacum* adaptation to cold stress (i.e. below its optimal growth temperature)?
- 8) Font sizes should be increased for better readability in several Figures (e.g. Fig. 2D, Fig. 3A, Fig. 4B, Fig. 4E...). All panels in Fig. 5 could be enlarged.

Reviewer #3 (Remarks to the Author):

In the manuscript, the authors describe the biochemical characterisation and crystal structure of two previously orphan enzymes that they found to be involved in the metabolism of intracellular polysaccharides in mycobacteria. This work is a significant contribution to a field of which strikingly very little is known. The authors have made two major discoveries: a) that the production of 3-O-methyl-mannose polysaccharides (MMPs) is widespread across the genus *Mycobacterium*, as opposed to being restricted to a subset of species as is currently understood, and b) a previously unrecognised family of mannoside hydrolases, that the authors show to be involved in the degradation of these MMPs. Additionally, their characterisation and solution of the 3D-structure of an α -(1→4)-mannosyltransferase will be of great interest to the field of carbohydrate enzymology, because mannosidic α -(1→4)-linkages are rare in nature and very few proteins have been described to perform such reaction. Moreover, the fact that their analysis is framed in the context of a whole genus, exemplifies the strengths of abandoning the prevalent single-species perspective. In my opinion, there are two aspects that would benefit from additional experimental evidence to support their conclusions: a) a more in-depth structural analysis (e.g. NMR, hydrolysis analysis) of the MMPs isolated from *M. avium* and b) a better quality characterisation of the mannoside hydrolase knock-out strain (e.g. complete growth curves at more than three different temperatures).

Other comments as follows:

1. Line 36. The connotation of "unique" does not seem clear to me and I think it is important. Unique to mycobacteria? Of unique in its catalytic activity? Or perhaps that it has not been described before?
2. Line 124. Figure 1B. *Hoyosella* and *Mycobacterium* are sister taxa, but this relationship is not reflected in the cladogram where *Hoyosella* wrongly appears to be more related to *Streptomyces*. Could I also suggest that the authors experiment with a different way to annotate this figure? I found the legends difficult to follow and not colour-blind friendly.
3. Line 148. Could the authors comment on the reason the MMPs from *M. avium* (Figure S1E) have such a different ionisation profile than the others, (e.g. re predominance of potassiated adducts).
4. Lines 220 and 276. Could the authors include in their supplementary information the alignment used to generate figure 3, in order to support the statement that the residues mentioned in the text are highly conserved among Actinobacteria.
5. Line 229. The reader would benefit from having an annotated (i.e. highlighting the residues mentioned in the text) structure-based sequence alignment of MmpH, G1d and CpGH125 (equivalent to figure S6B; perhaps instead of figure S2B).
6. Line 240. From the placing of reference 18 in the text, the reader infers that it is CAZy that suggests that families GH15 and GH125 are in the same clan, which it does not.
7. Line 276. Given that they are describing a novel family of proteins, could authors develop the comparison between family members across Actinobacteria. Is this family Actinobacteria specific?
8. Line 278. The authors refer specifically to the *M. hassiacum* gene cluster, but presumably the arrangement of the cluster is the same in all mycobacteria?
9. Line 293. Could figure 4B be bigger? I need to zoom to 300% in order to see the positions of the methylations of the synthesised compounds.
10. Line 355. Are not these three enzymes in the same CAZy family GT4? So why jump to the superfamily level?
11. Consistency of colour coding in figures 3 and 5 compared to the others. It would help the reader if MmpH was yellow in figure 3, and ManT green in figure 5.
12. Could SAM/SAH and GDP-Mannose/GDP be added to figure 7 to aid in the understanding of the catalytic activities of each of the enzymes.
13. I suggest to mention somewhere in the text or figures the Ordered Locus Names/ of the corresponding *M. smegmatis* MC² 155 and *M. avium* genes (MSMEG_NNNN, MAV_NNNN), in order to make their discoveries easily accessible to a wider audience.

Reviewer #1 (Remarks to the Author):

In this manuscript, Maranha et al. reported the novel recycling biosynthetic mechanism of 3-O-methylmannose polysaccharide (MMP) in nontuberculous mycobacteria (NTM). The authors' group previously revealed the structure and enzymatic features of MeT1 which is one of the key enzymes in the MMP biosynthetic pathway. They here characterized the structures and enzymatic properties of other key two enzymes, MMP hydrolase (MmpH) and mannosyltransferase (ManT). Most strikingly, the authors found that the products of MmpH were used as substrates for ManT and mannose 1-O-methyltransferase (MeT1), serving as a key for this recycling pathway. Overall, the study was well conducted, and the paper is well written. After several concerns will be addressed, I am happy to support publication of this study in *Commun Biol*. Specific comments are as follows.

Major comments

1. Regarding the structure of MmpH, even without ligand-complex crystal, the authors nicely discussed and predicted the substrate recognition mechanism by comparing with structurally related G1d and CpGH125. Such prediction should be experimentally confirmed using the mutant enzymes which lack any key putative residue.

We produced MmpH variants Asp47Ala, Asp50Ala and Glu262Ala (lines 605-607; 628-629) and assessed their activity, as requested, in order to confirm the role of these residues in the activity of the enzyme. The resulting data have been included in the revised manuscript (please see section "Overall structure of MmpH" [lines 250-261] and the two new panels [J and K] of Figure 2).

2. The authors detected sequential addition of up to 3 Man residues by ManT reaction in vitro. What would be the number of Man residues with longer incubation time or excess amount of the enzyme? This is an important point to estimate the biosynthetic mechanism of MMP.

We have performed an additional experiment to address this question and found that, even after longer incubation times, no higher order oligomannosides could be detected (please see the new panel D added to Fig. 4). A possible explanation would be that extension to a higher order oligomannoside requires participation of MeT3 (likely encoded by the *pmeT3* gene in the MMP operon). Two models have so far been proposed for MMP elongation, one by alternating and interdependent activities of a mannosyltransferase and a methyltransferase (Weismann and Ballou (1984) [https://doi.org/10.1016/S0021-9258\(17\)43116-4](https://doi.org/10.1016/S0021-9258(17)43116-4)), and another in which methylation would be independent of mannosylation (Xia and Lowary (2012) <https://doi.org/10.1002/cbic.201200121>). Our results are compatible with the activity of a 3-O-methyltransferase being required for full MMP polymerization by allowing extension of higher order unmethylated mannosides, that is, a hybrid model between those previously proposed by Ballou and Lowary. This, however, remains to be experimentally demonstrated and is a subject for further investigation.

3. Line 192: The authors described that MmpH showed the higher apparent K_m value at 45 degree than 37 degree. The higher V_{max} is well consistent with Fig. 2E, but I feel that the K_m values are only slightly different and very similar between 45 and 37. What do the authors want to claim from the "higher apparent K_m "?

Thank you for pointing out the similar K_m values. We have amended the text that now reads "comparable apparent K_m and higher calculated V_{max} " (please see line 196).

4. Although MmpH and ManT were the enzymes in the same pathway, Fig. 2 and 4 showed that the two recombinant enzymes derived from the same species have different optimal pH and degree. This should be discussed more carefully.

Although the pH and temperature profiles of MmpH and ManT are not very different, it is not uncommon to find differences between enzymes in the same organism and/or in the same biochemical pathway. Having similar physicochemical properties *in vitro* is not a requirement for enzymes of any given pathway to be able to perform their biological function. Indeed, what we designate "optimal conditions" for an enzyme *in vitro*, a necessarily simplified environment, can differ significantly from the conditions under which that enzyme will operate *in vivo*.

5. Introduction. Previous findings about the MMP biosynthetic pathway are nicely summarized, but information about the putative 3-O-methyltransferase (pMeT3) is lacking. I would suggest that the putative roles or related information of pMeT3 are also described.

We have included in the Introduction section (Lines 93-100) of the revised manuscript a brief description of the two models proposed for MMP elongation by the Ballou and Lowary laboratories, which also include the putative role of the methyltransferase (pMeT3) activity detected in mycobacterial cell extracts (soluble and membrane fractions). Please see also the answer to comment 2, above.

Minor points

6. Line 277: A blank line better be inserted before the title.
7. Line 36: "endomannosidade" should be "endomannosidase"
8. Line 174: "purify" should be corrected.
9. Line 181-184: "c" and "d" should be bold.
10. Line 297-298: "a" and "d" should be bold.

All minor points (6 to 10) have been addressed as suggested.

Reviewer #2 (Remarks to the Author):

The manuscript entitled "Self-recycling and partially conservative replication of mycobacterial methylmannose polysaccharides" by Maranhã, Costa, Ripoll-Rozada and colleagues studied the mycobacterial pathway involved in recycling and replication of 3-O-methylmannose polysaccharide (MMP). Using bioinformatics and phylogenetics, the authors describe the distribution of MMP biosynthetic gene clusters in mycobacterial genomes and found a wide distribution of these clusters in nontuberculous mycobacteria (NTM), including the first description of such genes in slowly growing organisms. They also biochemically and structurally characterized two enzymes important for MMP biosynthesis, an endo- α -(1 \rightarrow 4)-mannosidase that hydrolyses MMP and an α -(1 \rightarrow 4)-mannosyltransferase, the first protein described to be able to elongate a polysaccharide polymer with α -(1 \rightarrow 4)-linked mannoses. The endomannosidase products can be used as substrates for elongation of new MMP chains by the mannosyltransferase and by a methyltransferase coded by the same gene cluster. This study also provides evidence that MMP could be linked to mycobacterial cold adaptation. The authors conclude their study proposing an elegant mechanism of MMP biogenesis relying on pre-existing mature MMP and on the concerted action of the endomannosidase, the mannosyltransferase and two additional specific methyltransferases. In summary, the manuscript is a valuable contribution to the literature of biogenesis and recycling of mycobacterial polysaccharides.

The manuscript is well written, the work is technically sound and may be published after minor revisions addressing the points below:

- 1) All phylogenetic trees do not present bootstrap values. Please provide bootstrap or other branch support value to the phylogenetic trees.

Bootstrap measures the reliability of the internodes of cladograms generated by maximum likelihood analysis. Often, measurements for some branches will be <100 and the estimated phylogeny measured by the repeated sampling is distilled into a single “best-tree” that attempts to summarize branch support and capture the most frequently occurring phylogenetic relationships. Hence, trees resulting from maximum likelihood bootstrap analyses may have some different branches (when bootstrap values <100) and the “best-tree” is more informative. We therefore decided to keep the current tree in Fig. 1 with a clearer figure legend, and included, as example, one of the trees containing bootstrap values as new supplemental information (Fig. S1). The trees in Fig. S3 and Fig. S8 that were constructed using the Neighbor-Joining and Poisson correction methods, now include bootstrap values as requested.

2) The authors searched mycobacterial genomes looking for a 4-gene cluster involved in MMP biosynthesis, indicating extensive and scattered distribution of these genes across mycobacteria. But the fact that this cluster is also found in *Gordonia rubripertincta* and three *Nocardia* (*N. farcinica*, *N. otitidiscaviarum* and *N. pseudobrasiliensis*; Fig. 1B) suggests the distribution is even more widespread than the authors currently describe in the manuscript and the same pattern of extensive and scattered distribution could happen at least in the order *Corynebacteriales*. It would be interesting to analyze the broader distribution of MMP clusters in *Corynebacteriales* / *Actinomycetia*.

We agree that it would be interesting to analyze the distribution of these genes in all *Actinobacteria*, but in this particular study we have decided to restrict the analysis to mycobacteria. Nevertheless, we included in the phylogenetic tree some actinobacterial taxa, closely related to mycobacteria and in which the polysaccharides had been analyzed, and where we have now confirmed the presence of this the gene cluster. Additionally, we mentioned in the Fig. 1 legend that a similar acylated MMP had been isolated from *Streptomyces griseus*, again suggesting that the MMP gene cluster has broader distribution within the *Actinobacteria* and is not restricted to mycobacteria.

3) There are two references to “data not shown” in the manuscript (lines 160 and 310). All relevant data should be shown and provided in the manuscript (e.g. as supplementary material) or references to these data should be entirely removed from the study.

For the first reference to “data not shown”, we have included a new panel A in former Fig. S3 (Fig. S4A in the revised manuscript) showing the absence of MmpH activity with MGLP or its deacylated form, β -mannans or synthetic 4 α -oligomannosides. The second instance was removed from the manuscript.

4) The manuscript nicely describes specific mutations affecting the activity of ManT, but not of MmpH. There are a few positions that seem to be essential for the MmpH function, such as D47, D50, E262, and residues lining a potential subsite occupied by glycerol. The confirmation of which residues are important for the catalytic activity and sugar binding of MmpH would strengthen the work.

We have produced the suggested MmpH variants and confirmed the functional role of these residues. This information has been included in the revised manuscript, as suggested. Please refer to the response to Major comment 1 from Reviewer 1, above, for further details. Also, a description of the construction of ManT variants that was missing from the original text has now been included (lines 641-643).

5) It is interesting to notice that the ManT E313A mutant, although having lower enzymatic activity (Fig. 5E), seems to be able to produce sP1 (pentamannoside) and sP2 (hexamannoside) products, but not sP3 (heptamannoside), in contrast to the Wt protein (Fig. 5F). What would be an explanation for this inability of the E313A mutant to produce heptamannosides?

We suspect that the apparent absence of heptamannoside product might result from the lower enzymatic activity of the ManT Glu313Ala variant. Indeed, for the wild-type enzyme the heptamannoside product is much less abundant than the hexamannoside, and it is likely below detection level for the variant enzyme. In any case, we have not delved into this question further.

6) Conservation analysis and electrostatic surface potential could be mapped onto the ManT structure, similarly to the analyses shown in Figures 3C-D for MmpH.

Two new panels (E and F) were added to Fig. 5, as suggested, and the text was amended accordingly (lines 370-373). Following the request by Reviewer 3 to include the alignment used to generate Fig. 3C, for consistency we have also incorporated the ConSurf alignment for ManT as supplementary data in Fig. S10.

7) The source of the enzymes studied in this work is *M. hassiacum*, a thermophilic organism that grows optimally at 50 °C and up to 65 °C. However, the optimal activities of ManT and MmpH are in the 40 - 45 °C range. The manuscript provides clues linking MMP to cold adaptation; is it possible that there is a relationship between the smaller thermostability of these enzymes and *M. hassiacum* adaptation to cold stress (i.e. below its optimal growth temperature)?

At this stage, we do not establish a direct relationship between enzyme maximal activity at 40-45°C (slightly below 50°C, the optimum for growth) and the role of MMP in cold adaptation. Optimal enzyme activity *in vitro*, a simplified environment where the enzymes lack their cellular interactors, may not mirror optimal activity *in vivo* and is instead an approximate indicator. However, we did not examine thermostability, i.e., the residual activity of the enzyme after exposure to a certain temperature during a period of time.

8) Font sizes should be increased for better readability in several Figures (e.g. Fig. 2D, Fig. 3A, Fig. 4B, Fig. 4E...). All panels in Fig. 5 could be enlarged.

We increased font sizes in the figures and the size of Fig. 5 panels, as suggested.

Reviewer #3 (Remarks to the Author):

In the manuscript, the authors describe the biochemical characterisation and crystal structure of two previously orphan enzymes that they found to be involved in the metabolism of intracellular polysaccharides in mycobacteria. This work is a significant contribution to a field of which strikingly very little is known. The authors have made two major discoveries: a) that the production of 3-O-methyl-mannose polysaccharides (MMPs) is widespread across the genus *Mycobacterium*, as opposed to being restricted to a subset of species as is currently understood, and b) a previously unrecognised family of mannoside hydrolases, that the authors show to be involved in the degradation of these MMPs. Additionally, their characterisation and solution of the 3D-structure of an α -(1→4)-mannosyltransferase will be of great interest to the field of carbohydrate enzymology, because mannosidic α -(1→4)-linkages are rare in nature and very few proteins have been described to perform such reaction. Moreover, the fact that their analysis is framed in the context of a whole genus, exemplifies the strengths of abandoning the prevalent single-species perspective. In my opinion, there are two aspects that would benefit from additional experimental evidence to support their conclusions: a) a more in-depth structural analysis (e.g. NMR, hydrolysis analysis) of the MMPs isolated from *M. avium* and b) a better quality characterisation of the mannoside hydrolase knock-out strain (e.g. complete growth curves at more than three different temperatures).

We concur with the reviewer that it would be interesting to perform the assays suggested. Unfortunately, not only is *M. avium* far from being a growth-friendly bacterium, but also the purification of MMP from this source was a rather laborious and time-consuming procedure, which we carried out nevertheless to demonstrate that this slow-growing mycobacterium also produced MMP, once thought to be absent from this group of mycobacteria. Further, the MMP purified from the fast-growers *M. smegmatis* and *M. hassiacum* was a heterogeneous mixture (11-14-mers), which further complicates an exhaustive structural characterization. The optimization of the production and purification of *M. avium* MMP to obtain considerable amounts of pure, homogeneous material and its structural characterization are therefore unfeasible in the timeframe of a few months. Still, our goal was served as we overturned an old paradigm. As for the curves with the MMP-negative mutant at more than 3 different temperatures, it is for us unclear how those would provide additional insight, compared to the growth data provided. Our data showed that the *M. smegmatis* mutant lacking MMP could not maintain the growth rate at 15°C that it exhibited at higher temperatures. Since this temperature approximates the average ambient temperatures of the natural environment of NTM and of public water distribution systems, we believe we have opened the way for future studies that explore this MMP-cold connection, maybe including its role in their ability to form biofilms, in their survival rates in dormant form, and even in their resistance to freezing.

Other comments as follows:

1. Line 36. The connotation of “unique” does not seem clear to me and I think it is important. Unique to mycobacteria? Of unique in its catalytic activity? Or perhaps that it has not been described before?

“Unique” was used to highlight that this catalytic activity had not been described before. We have rephrased this sentence in order to make it clearer (line 36).

2. Line 124. Figure 1B. *Hoyosella* and *Mycobacterium* are sister taxa, but this relationship is not reflected in the cladogram where *Hoyosella* wrongly appears to be more related to *Streptomyces*. Could I also suggest that the authors experiment with a different way to annotate this figure? I found the legends difficult to follow and not colour-blind friendly.

The original description and phylogenetic position of the genus *Hoyosella* was determined based on the 16s rRNA gene (<https://doi.org/10.1099/ijs.0.008664-0>). Subsequent studies describe novel species of *Hoyosella* based on phenotypic, chemotaxonomic and phylogenetic analysis supported on the 16s rRNA gene, as well as on Average nucleotide identity (ANI) and digital DNA-DNA hybridization (dDDH). In *Mycobacterium*, the phylogenetic position based on the 16s rRNA gene alone has little resolution and, to better define species, two other housekeeping genes have been used. The BCGTree tool selected to reconstruct the phylogenetic tree in Fig. 1 uses 107 essential single copy core genes, which allows the phylogeny of bacterial strains to be resolved in more detail (<https://doi.org/10.1139/gen-2015-0175>). Not excluding the possibility of bias resulting from the much higher number of *Mycobacterium* genomes in comparison to those of other genera, the position of *Hoyosella* was generated with the BCGTree bioinformatic tool. We would therefore like to keep the tree in Fig. 1. Also, the legend of Fig. 1 was simplified and, wherever possible, labeled in a color-blind friendly manner, as requested.

3. Line 148. Could the authors comment on the reason the MMPs from *M. avium* (Figure S1E) have such a different ionisation profile than the others, (e.g. re predominance of potassiated adducts).

When analyzed in positive ion mode with electrospray ionization, sugars tend to interact with positive ions (such as sodium or potassium) and form adducts (J Am Soc Mass Spectrom. 2007;18(2):332-6. <https://doi.org/10.1016/j.jasms.2006.10.002>). In this work, the source of positive ions for such an interaction is most likely the culture media in which the bacteria were

grown. After bacterial growth, MMP polysaccharides for Mass Spectrometry analysis were extracted using two different processes (as described in the Methods section). *M. smegmatis* MMP (Fig. S2B in the revised manuscript), was extracted from 30 g of cell biomass, with organic solvents in a rotary evaporator, and purified in two chromatographic steps (reversed-phase chromatography on a Resource RPC column followed by size-exclusion chromatography on a HiPrep 16/60 Sephacryl S-200 HR column), whereas *M. avium* MMP (Fig. S2E in the revised manuscript), was extracted and purified using a miniaturized protocol, departing from much less (4 mL or a loopful) bacterial biomass, solvent-extracted in two consecutive steps, followed by solid-phase extraction on a 3 mL HyperSep C18 SPE cartridge. Given that the samples were analyzed by electrospray ionization direct infusion in positive ion mode, the simpler purification process used for *M. avium* MMP could account for a higher concentration of potassium ions, and hence a different ionization profile with predominance of potassium adducts.

4. Lines 220 and 276. Could the authors include in their supplementary information the alignment used to generate figure 3, in order to support the statement that the residues mentioned in the text are highly conserved among Actinobacteria.

The ConSurf alignment used to generate Fig. 3 has now been included as supplementary data (Fig. S7). The ConSurf alignment used for conservation analysis of ManT structure (Fig. 5E-F, requested by Reviewer 2) has now been included as Fig. S10. An additional amino acid sequence alignment has also been added to Fig. S3 (panel B) to provide support to the claims of high conservation of several residues among Actinobacteria (please see also the answers to comments 5 and 7, below).

5. Line 229. The reader would benefit from having an annotated (i.e. highlighting the residues mentioned in the text) structure-based sequence alignment of MmpH, G1d and CpGH125 (equivalent to figure S6B; perhaps instead of figure S2B).

We agree that a structure-based sequence alignment can be very useful to help understanding these questions. Unfortunately, in this particular case, the very low sequence identity and lack of conservation of the relative positions of some of the structurally overlapping elements along the sequences (e.g., Asp47 of MmpH corresponds to Asp220 of CpGH125 and Trp315 of MmpH corresponds to Trp62 of CpGH125), make it impossible to provide a meaningful sequence-based alignment. In any case, we have replaced Fig. S2B (now Fig. S3B) with an alignment of MmpH orthologues that also highlights the high conservation among Actinobacteria of some residues identified in the text.

6. Line 240. From the placing of reference 18 in the text, the reader infers that it is CAZy that suggests that families GH15 and GH125 are in the same clan, which it does not.

Indeed, reference 18 was misplaced here and it has been eliminated from this line but maintained wherever appropriate across the manuscript to acknowledge the CAZy database.

7. Line 276. Given that they are describing a novel family of proteins, could authors develop the comparison between family members across Actinobacteria. Is this family Actinobacteria specific?

The claim in the text that the binding site residues are conserved across Actinobacteria is now supported by an alignment added to Fig. S3B (previously S2B; cf. answer to comment 4, above). Sequence comparison analysis with BLAST reveals relevant sequence identity (40-51%) between *M. hassiacum* MmpH and some non-actinobacterial proteins (namely in some Proteobacteria). There is also considerable conservation of binding site residues. However, further extending the comparison outside Actinobacteria, falls outside the objectives of this study.

8. Line 278. The authors refer specifically to the *M. hassiacum* gene cluster, but presumably the arrangement of the cluster is the same in all mycobacteria?

In most *Mycobacterium* species (>90%) the genetic arrangement is the same as that in *M. hassiacum*. There are however some exceptions where *met1* is not immediately adjacent to *mmpH* (*M. gadium* JCM 12688; *M. arabiense* JCM 18538; *M. madagascariense* JCM 13574; *M. alvei* JCM 12272; *M. lutetiense* strain DSM 46713; *M. sediminis* JCM 17899; *M. insubricum* JCM 16366) and there are cases where the distance between *met1* and *mmpH* varies from genome to genome. We have created Fig. S1B to illustrate these differences.

9. Line 293. Could figure 4B be bigger? I need to zoom to 300% in order to see the positions of the methylations of the synthesised compounds.

Figure 4B was enlarged as requested.

10. Line 355. Are not these three enzymes in the same CAZy family GT4? So why jump to the superfamily level?

We agree with the reviewer and the sentence now reads “Despite low amino acid sequence conservation (Fig. S8B), the three enzymes belong to the GT4 family and display a GT-B fold (Fig. 5B)”.

11. Consistency of colour coding in figures 3 and 5 compared to the others. It would help the reader if MmpH was yellow in figure 3, and ManT green in figure 5.

The color codes were altered as requested.

12. Could SAM/SAH and GDP-Mannose/GDP be added to figure 7 to aid in the understanding of the catalytic activities of each of the enzymes.

We are grateful for the suggestion to improve the figure and performed the requested additions.

13. I suggest to mention somewhere in the text or figures the Ordered Locus Names/ of the corresponding *M. smegmatis* MC² 155 and *M. avium* genes (MSMEG_NNNN, MAV_NNNN), in order to make their discoveries easily accessible to a wider audience.

The codes were added for the *M. hassiacum*, *M. smegmatis* and *M. avium* genes (lines 525 – 529).

REVIEWERS' COMMENTS:

Reviewer #1 (Remarks to the Author):

The authors have adequately addressed all my concerns. I think that the paper is now ready for publication.

Reviewer #2 (Remarks to the Author):

As previously stated, this manuscript is a valuable contribution to the literature of biogenesis and recycling of mycobacterial polysaccharides. It is well written, the work is technically sound and as most of the suggestions from reviewers have been addressed, I am happy to support publication of this study in Communications Biology.

Reviewer #3 (Remarks to the Author):

In my opinion, the revised version of this manuscript has been strengthened by the additional experiments, clearer discussion and figure additions and modifications. The authors have adequately addressed my comments and concerns.

As well as having seen some typos in the methods (e. g. should it not be QuikChange in line 606 and wild-type gene in line 643?), I have two comments regarding the cladograms. Shouldn't *M. kansasii* and closely related species be red in figure 1B? and what is the difference between the cladogram in S1A vs the one in 1B?

REVIEWERS' COMMENTS:

Reviewer #1 (Remarks to the Author):

The authors have adequately addressed all my concerns. I think that the paper is now ready for publication.

Reviewer #2 (Remarks to the Author):

As previously stated, this manuscript is a valuable contribution to the literature of biogenesis and recycling of mycobacterial polysaccharides. It is well written, the work is technically sound and as most of the suggestions from reviewers have been addressed, I am happy to support publication of this study in Communications Biology.

Reviewer #3 (Remarks to the Author):

In my opinion, the revised version of this manuscript has been strengthened by the additional experiments, clearer discussion and figure additions and modifications. The authors have adequately addressed my comments and concerns.

As well as having seen some typos in the methods (e. g. should it not be QuikChange in line 606 and wild-type gene in line 643?), I have two comments regarding the cladograms. Shouldn't *M. kansasii* and closely related species be red in figure 1B? and what is the difference between the cladogram in S1A vs the one in 1B?

We are grateful to the three Reviewers for their rigorous evaluation of our manuscript. Naturally, we have made the necessary corrections to all the mistakes accurately detected by Reviewer #3. Indeed, the uploaded Figure 1 contained a precursor cladogram (panel B) still with bootstrap values and not the intended "Best Tree", a difference explained to Reviewer #2 upon his/her request in the previous Rebuttal Letter. In such cladogram, *M. kansasii* and 5 other closely related species had not yet been marked red (no significant homology to the MMP cluster detected), as duly noted by Reviewer #3.